# An electronic health records cohort study on heart failure following myocardial infarction in England: incidence and predictors

Johannes M I H Gho,[1,2] Amand F Schmidt,[2,3,4] Laura Pasea,[1] Stefan Koudstaal,[1,2] Mar Pujades-Rodriguez,[1,5] Spiros Denaxas,[1] Anoop D Shah,[1] Riyaz S Patel,[1,3] Chris P Gale,[6] Arno W Hoes,[7] John G Cleland,[8] Harry Hemingway,[1] Folkert W Asselbergs[1,2,3,9]

## ABSTRACT

**Objectives** To investigate the incidence and determinants of heart failure (HF) following a myocardial infarction (MI) in a contemporary cohort of patients with MI using routinely collected primary and hospital care electronic health records (EHRs).

**Methods** Data were used from the CALIBER programme, linking EHRs in England from primary care, hospital admissions, an MI registry and mortality data. Subjects were eligible if they were 18 years or older, did not have a history of HF and survived a first MI. Factors associated with time to HF were examined using Cox proportional hazard models.

**Results** Of the 24 479 patients with MI, 5775 (23.6%) developed HF during a median follow-up of 3.7 years (incidence rate per 1000 person-years: 63.8, 95% CI 62.2 to 65.5). Baseline characteristics significantly associated with developing HF were: atrial fibrillation (HR 1.62, 95% CI 1.51 to 1.75), age (per 10 years increase: 1.45, 1.41 to 1.49), diabetes (1.45, 1.35 to 1.56), peripheral arterial disease (1.38, 1.26 to 1.51), chronic obstructive pulmonary disease (1.28, 1.17 to 1.40), greater socioeconomic deprivation (5th vs 1st quintile: 1.27, 1.13 to 1.41), ST-segment elevation MI at presentation (1.19, 1.11 to 1.27) and hypertension (1.16, 1.09 to 1.23). Results were robust to various sensitivity analyses such as competing risk analysis and multiple imputation.

**Conclusion** In England, one in four survivors of a first MI develop HF within 4 years. This contemporary study demonstrates that patients with MI are at considerable risk of HF. Baseline patient characteristics associated with time until HF were identified, which may be used to target preventive strategies.

## Strengths and limitations of this study

► This study based on the use of linked electronic health records from general practitioners and hospital records describes the current burden of heart failure (HF) in a representative sample of patients with a first myocardial infarction.

► The linkage of data from three sources (disease registry, primary care and hospital records) improved diagnostic ascertainment and accuracy in timing of events.

► Misclassification of drug exposure was likely to be minimal, as prescriptions issued during consultation are automatically recorded.

► Risk factor adjustment might have been incomplete given that information regarding baseline body mass index, smoking and blood pressure was missing for 34% to up to 70% of patients. Due to the high degree of missing data on time to revascularisation (88.3%), we did not explore its relation with HF incidence.

► Stratified methods were used to account for potential calendar and centre effects, and competing risk models were used to adjust for potential competing effects on HF and mortality.

## INTRODUCTION

Research describing the incidence of heart failure (HF) following myocardial infarction (MI) is limited, mainly originating from the thrombolytic era, often using small sample sizes with contradictory findings, making it difficult to provide evidence-based medicine. For example, a previous UK study among almost 900 patients hospitalised with MI in 1998 found that one-fifth developed HF during their hospital stay and a further third following hospital discharge.[1] More recently, a Swedish study found a 5-year cumulative risk of HF after MI of 21.8% in the calendar period 2004–2013.[2] Further, a Danish nationwide cohort study reported an incidence of HF at 90 days following MI of 19.6% in 2009–2010.[3] Differences between these studies could be related to a number of factors, for example, change in treatment, national policies or definitions of HF, all of which potentially limit the generalisability of results. We used a large contemporary and representative

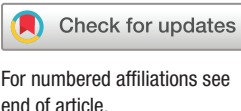

For numbered affiliations see end of article.

**Correspondence to**
Dr Amand F Schmidt;
amand.schmidt@ucl.ac.uk

sample of patients with MI based on electronic health records (EHRs) to (1) describe the current incidence of HF after MI in England, using both primary and hospital care data, and (2) explore patient characteristics predictive of post-MI HF.

## METHODS

### Study design

Data were used from the CALIBER dataset, which included linked data from (1) primary care EHRs, with diagnoses coded using the Read terminology (Clinical Practice Research Datalink, CPRD) from 226 consenting general practices,[4] (2) hospital care administrative records (Hospital Episode Statistics, HES) and disease registry/audit records (Myocardial Ischaemia National Audit Project, MINAP) and (3) the mortality register (from UK Office for National Statistics, ONS).[5]

### Setting and participants

Patients were eligible if they were aged ≥18 years, registered in CPRD practices in England consenting to data linkage, with at least 1 year of up-to-standard prestudy follow-up (meeting CPRD practice quality standards) and experiencing a first MI between 1 January 1998 and 25 March 2010. The first record of MI across the linked data sources was considered as the index event, and subsequent MI records within 30 days from the same or other sources were considered as representing the same event.[6] Patients with a fatal index MI or a history of HF before their index MI were excluded. Patients were censored if they ceased to be registered at a participating general practice, at the date of death or the administrative censoring of the dataset (25 March 2010).

### Variables and data sources

Study variables were derived from diagnoses recorded across several controlled clinical terminology and statistical classification systems: Read, International Classification of Diseases (ICD)-9 or ICD-10, medication prescription information or the Office of Population Censuses and Surveys (OPCS) Classification of Interventions and Procedures-4 codes (see https://rdrr.io/rforge/CALIBERcodelists/man/PRODDICT.html) with the EHR phenotyping algorithms published (https://caliberresearch.org/portal).[5]

For continuous variables, the most recent measurement recorded in CPRD in the year before study entry was used as the baseline value. Data before study entry were used to determine the prognostic potential of age (years, using the date of birth), sex, ethnicity, social deprivation (Index of Multiple Deprivation, IMD score as recorded in ONS in quintiles), smoking, alcohol use, history of cardiovascular disease (CVD), previous coronary revascularisation, history of diabetes, history of thyroid disease, history of chronic obstructive pulmonary disease (COPD), history of depression and history of non-metastatic cancer (see the CALIBER portal (https://

caliberresearch.org/portal) for details). Diabetes and hypertension diagnosis were based on Read codes from primary care or ICD-10 codes from HES, both of which were classified using primary care consultation records and hospital diagnosis records. Socioeconomic status was based on the IMD which includes seven domains of deprivation (income deprivation, employment deprivation, health deprivation and disability, education, skills and training deprivation, barriers to housing and services and living environment deprivation and crime).[7] The index MI event was characterised using ECG findings (eg, ST-segment elevation myocardial infarction (STEMI) or non-STEMI (NSTEMI)), site of infarction, mode and timing of reperfusion and peak cardiac biomarkers (troponin I, troponin T and creatine kinase). MI was defined by linking all of the CALIBER data sources (eg, HES and CPRD) with the type of MI recorded in MINAP and CPRD, see for more detail https://www.caliberresearch.org and the appendix. Characteristics regarding the index MI (eg, type of revascularisation and delay from symptom onset to delivery of reperfusion therapy) were derived from MINAP. This nationwide registry is part of an annual audit which records over 20 key MI variables.[8] Herrett and colleagues[6] validated the current approach and showed the necessity of linking multiple sources to ascertain MI (see appendix for codes used). The primary study outcome was the first HF event following MI, which was similar to MI recorded in multiple CALIBER data sources; echocardiographic findings and New York Heart Association class were unavailable. All phenotypes were created and validated using a robust methodology described elsewhere[9] and have been used in previously published studies.[10 11]

### Statistical methods

Incidence rates (cases per 1000 person-years) and Kaplan-Meier cumulative incidence rates of HF were estimated, with Kaplan-Meier curves stratified by age (<50, 50–65 and ≥65 years) and type of MI. Association between baseline variables with the onset of HF following MI was explored using Cox proportional hazard models. The proportional hazards assumption was checked using Schoenfeld residuals and by non-parametric correlation coefficients between survival time and the parameter specific residuals. All models were stratified on general practice and calendar year periods of enrolment (1998–2001, 2001–2004, 2004–2007 and 2007–2010). Stratified models were used instead of frailty models because the former does not make any distributional assumption. Models were sequentially adjusted for: (1) age and sex, (2) cardiovascular risk factors, (3) type of MI and (4) comorbidities and prescribed medication. Associations are presented as HRs with 95% CIs. Analyses were performed using R (V.3.3.2).[12]

### Sensitivity analyses

Due to the availability of EHR from multiple sources, clinical diagnoses and prescriptions were completely

recorded. Biomarker and lifestyle measurements such as smoking, body mass index (BMI) and blood pressure were, however, incompletely recorded (see table 1). It is probable that these data were preferentially recorded in subjects perceived to be at a higher risk for early progression of CVD. Pairwise analyses regressing a complete case indicator on observed variables indeed showed considerable dependency between recorded data and missingness (results not shown), violating the 'missing completely at random' assumption. This dependency was used by multiply imputing missing data using the mice package, which was implemented using 20 imputed datasets and pooled based on Rubin's rules (online supplementary methods).[13]

To account for the fact that patients may have died before the onset of HF (eg, competing risk by mortality), analyses were repeated using Fine and Gray models.[14 15] As a hypothetical example of competing risk, let us assume that out of 100 subjects with MI, 1 develops HF and 99 die. In a Cox's regression analysis, all 99 dead subjects would be censored (implicitly, and incorrectly, assuming they may develop HF later in time), and the hazard of HF would equal 1/1. Instead, a Fine and Gray model recognises that the 99 dead subjects could never develop HF and hence calculates the hazard as 1/100.

In addition to these sensitivity analyses, we performed a cancer-stratified and revascularisation-stratified analysis. Due to discrepancies in sample size between the subgroups, we decided against formal interaction testing, which may suffer from inflated false positive rates and lower power in such settings.[16]

## RESULTS
### Subjects and baseline characteristics
In total, there were 52 770 patients with an index MI during the study period (figure 1). Excluding patients with a fatal index MI (n=15 104), a prior history of HF (n=3876), missing information on the type of MI (n=97) and subjects with less than 1 year of follow-up prior to the indexing MI event (n=9214) resulted in an analytical cohort of 24 479 patients. Patients were recruited from 226 general practitioner (GP) practices, with the median practice enrolling 101 patients (quartile (Q1) 65 and Q3 150). Of the included subjects, 4657 patients were STEMI and 19 822 patients were NSTEMI (0% missing), 15 969 (65.2%) were men (0% missing), 4538 (42.1%) currently smoked (56% missing), 12 258 (50.1%) had hypertension (0% missing) and 3014 (12.3%) had a history of diabetes at baseline (0% missing). In total, 6129 (25.0%) were prescribed beta-blockers (0% missing) and 5039 (20.6%) ACE inhibitors (0% missing) prior to their indexing MI (table 1). The number of patients identified across the previously defined calendar periods was reasonably stable, with a minimum of 5219 subjects in the period 1998–2001 compared with a maximum of 6838 subjects identified in the period 2004–2007.

### Incidence
Patients contributed 90 482 person-years of follow-up, during a median follow-up time of 3.7 years (IQR 1.5; 6.7), and 5775 patients (23.6%) developed HF. The crude incidence rate of HF following a first MI was 63.8 (95% CI 62.2 to 65.5) per 1000 person-years. Within the first 30 days post-MI follow-up, 2438 (10.0%) patients developed HF (figure 2). From day 30 onwards, 3337 (15.8%) of patients with MI (event free during the first 30 days) developed HF, with 6.8% experiencing an HF event within the first year (figure 2). The incidence of HF during the first 30 days of follow-up was 4.3% (102) in patients younger than 50 years, 6.0% (459) in 50–65 year olds and 12.9% (1877) in those 65 years and older, with HF incidence increasing proportionally as time progressed towards 10 years (figure 3 top row). The 30-day incidence of HF was 9.5% (1892 events) for subjects with NSTEMI and 11.7% (546 events) for subjects with STEMI. Excluding patients who experienced HF within the first 30 days showed that patients with STEMI had a lower incidence of HF than subjects with NSTEMI (figure 3 lower panels). At 57 days, the crude cumulative risk of HF in the subjects with NSTEMI surpassed that of the subjects with STEMI for the first time (0.0151 vs 0.0147), with the curves further diverging at 73 days of follow-up (since indexing MI).

Over the entire follow-up, 5921 subjects died, of which 3538 were free of HF at the time. During the first 30 days of follow-up, only one patient died, limiting the potential for competing risk by all-cause mortality. In patients who did not have HF after 30 days accounting for competing risk by all-cause mortality attenuated the cumulative risk to be about 15% after 10 years of follow-up (online supplementary data). Furthermore, the cumulative risks of all-cause mortality and HF converged as follow-up time progressed towards 10 years.

### Predictors of HF after multivariable adjustments
Table 2 describes the results of the previously described nested Cox's models, showing little change in estimates between the increasingly complex models. Focusing on the final model, the hazard of HF increased with atrial fibrillation (HR 1.62, 95% CI 1.51 to 1.75), age (per 10 years increase: 1.45, 1.41 to 1.49), diabetes (1.45, 1.35 to 1.56), peripheral arterial disease (1.38, 1.26 to 1.51), COPD (1.28, 1.17 to 1.40), hypertension (1.16, 1.09 to 1.23), higher socioeconomic deprivation (P value for the five groups <0.001) and STEMI at presentation (1.19, 1.11 to 1.27). Accounting for competing risk by all-cause mortality had little impact on the presented results (online supplementary data). After multiply imputing, the data models were extended to include smoking, BMI and systolic and diastolic blood pressure variables which showed BMI, male sex and smoking to be conditionally independent prognostic factors (online supplementary data). The extended model 5 was implemented using imputed data and non-imputed (complete case) data, resulting in similar HR estimates (magnitude and direction). Stratifying the sample on cancer diagnosis

**Table 1** Baseline characteristics at index MI

| | Patients with STEMI n=4657 | Patients with NSTEMI n=19822 | Total of patients with MI n=24479 | Unknown (%) |
|---|---|---|---|---|
| Follow-up time (years), median (IQR) | 3.3 (1.4–5.7) | 3.9 (1.5–7.1) | 3.7 (1.5–6.7) | 0 |
| Mean age, years (SD) | 65.7 (13.2) | 68.7 (13.2) | 68.1 (13.2) | 0 |
| Male sex | 3311 (71.1%) | 12658 (63.9%) | 15969 (65.2%) | 0 |
| Ethnicity | | | | 4.8 |
| White | 3466 (76.4%) | 14288 (76.1%) | 17754 (76.2%) | |
| Asian | 83 (1.8%) | 327 (1.7%) | 410 (1.8%) | |
| Black | 20 (0.4%) | 63 (0.3%) | 83 (0.4%) | |
| Other | 968 (21.3%) | 4085 (21.7%) | 5053 (21.7%) | |
| Body mass index (kg/m$^2$) | | | | 70.0 |
| Underweight (<18.5) | 22 (1.7%) | 120 (2.0%) | 142 (1.9%) | |
| Normal (18.5–25) | 335 (25.5%) | 1661 (27.4%) | 1996 (27.2%) | |
| Overweight (25–30) | 557 (43.0%) | 2431 (40.2%) | 2988 (40.7%) | |
| Obese (>30) | 381 (29.4%) | 1842 (30.4%) | 2223 (30.2%) | |
| Index of Multiple Deprivation Most deprived quintile | 892 (19.2%) | 3985 (20.2%) | 4877 (20%) | 0.4 |
| Risk factors before index MI | | | | |
| Current smoker | 1235 (51.1%) | 3303 (39.5%) | 4538 (42.1%) | 55.9 |
| Excess alcohol consumption | 55 (9.0%) | 219 (8.1%) | 274 (8.3%) | 86.4 |
| History of atrial fibrillation | 358 (7.7%) | 2210 (11.1%) | 2568 (10.5%) | 0 |
| History of hypertension | 2117 (45.5%) | 10141 (51.2%) | 12258 (50.1%) | 0 |
| History of peripheral arterial disease | 236 (5.1%) | 1475 (7.4%) | 1711 (7%) | 0 |
| Previous revascularisation | | | | |
| PCI | 700 (15.0%) | 2044 (10.3%) | 2744 (11.2%) | 0 |
| CABG | 126 (2.7%) | 843 (4.3%) | 969 (4.0%) | 0 |
| Previous TIA | 154 (3.3%) | 998 (5.0%) | 1152 (4.7%) | 0 |
| Previous stroke | 80 (1.7%) | 410 (2.1%) | 490 (2.0%) | 0 |
| History of diabetes | 538 (11.6%) | 2476 (12.6%) | 3014 (12.3%) | 0 |
| History of thyroid disease | 237 (5.1%) | 1374 (6.9%) | 1661 (6.6%) | 0 |
| History of COPD | 304 (6.5%) | 1634 (8.2%) | 1938 (7.9%) | 0 |
| History of non-metastatic cancer | 478 (10.3%) | 2282 (11.5%) | 2760 (11.3%) | 0 |
| Vital signs before admission, median (IQR) | | | | |
| Systolic blood pressure, mm Hg | 140 (130–153) | 140 (130–154) | 140 (130–154) | 34.2 |
| Diastolic blood pressure, mm Hg | 80 (74–88) | 80 (71–88) | 80 (72–88) | 34.2 |
| Biomarkers, median (IQR) | | | | |
| Troponin I (maximum) | 18.4 (3.6–50.0) | 2.2 (0.3–10.0) | 3.88 (0.62–21.4) | 88.3 |
| Troponin T (maximum) | 1.8 (0.60–4.8) | 0.4 (0.14–1.1) | 0.65 (0.18–2) | 90.8 |
| CK (maximum) | 721 (222–1631) | 219 (97–648) | 331 (123–1068) | 79.8 |
| Biomarkers before index MI, mean (SD) | | | | |
| Haemoglobin, g/dL | 14.1 (1.72) | 13.6 (1.84) | 13.7 (1.83) | 67.1 |
| White cell count | 8.0 (2.94) | 7.9 (2.92) | 7.9 (2.92) | 69.1 |
| Neutrophil count | 5.0 (2.12) | 5.0 (2.40) | 5.0 (2.35) | 71.9 |
| Platelets | 270 (85.2) | 266 (88.0) | 267 (87.5) | 69.1 |
| Erythrocyte sedimentation rate | 19.5 (18.8) | 22.2 (21.6) | 21.8 (21.2) | 89.4 |
| Creatinine, µmol/L | 98.5 (39.1) | 102.5 (47.3) | 102 (46.0) | 58.7 |

Continued

**Table 1** Continued

| | Patients with STEMI n=4657 | Patients with NSTEMI n=19 822 | Total of patients with MI n=24 479 | Unknown (%) |
|---|---|---|---|---|
| eGFR-CKD-EPI | 70.4 (19.9) | 66.2 (19.9) | 67.0 (20.0) | 60.4 |
| Random glucose concentration, mmol/L | 7.09 (3.53) | 7.26 (3.89) | 7.23 (3.82) | 76.5 |
| Total cholesterol | 5.39 (1.45) | 5.24 (1.28) | 5.27 (1.31) | 63.4 |
| LDL cholesterol | 3.24 (1.12) | 3.08 (1.10) | 3.11 (1.11) | 81.3 |
| Revascularisation characteristics | | | | |
| Site of infarction Anterior | 393 (40.2%) | 255 (30.7%) | 648 (35.9%) | 92.6 |
| Primary PCI | 258 (19.3%) | 73 (3.2%) | 331 (9.2%) | 85.3 |
| Prehospital fibrinolysis | 324 (7.0%) | 94 (0.5%) | 418 (1.7%) | 0 |
| Median (IQR) delay from symptom to reperfusion (min) | 150 (98–280) | 160 (101–296) | 153 (99–285) | 88.3 |
| Prescribed medication before index MI | | | | |
| Antiplatelet | 867 (18.6%) | 6979 (35.2%) | 7846 (32.1%) | 0 |
| Oral anticoagulant | 67 (1.4%) | 624 (3.1%) | 691 (2.8%) | 0 |
| Statin | 825 (17.7%) | 5297 (26.7%) | 6122 (25.0%) | 0 |
| ACE inhibitor | 676 (14.5%) | 4363 (22.0%) | 5039 (20.6%) | 0 |
| Angiotensin receptor blocker | 220 (4.7%) | 1208 (6.1%) | 1428 (5.8%) | 0 |
| Beta-blocker | 766 (16.4%) | 5363 (27.1%) | 6129 (25.0%) | 0 |
| Calcium channel blocker | 804 (17.3%) | 4621 (23.3%) | 5425 (22.2%) | 0 |
| Loop diuretic | 255 (5.5%) | 2188 (11.0%) | 2443 (10.0%) | 0 |
| Aldosterone antagonist | 24 (0.5%) | 198 (1.0%) | 222 (0.9%) | 0 |
| Digoxin | 53 (1.1%) | 515 (2.6%) | 568 (2.3%) | 0 |

Prehospital and inhospital fibrinolysis are not mutually exclusive.

CABG, coronary artery bypass grafting; CK, creatine kinase; COPD, chronic obstructive pulmonary disease; eGFR-CKD-EPI, estimated glomerular filtration rate using CKD-EPI; LDL, low-density lipoprotein; MI, myocardial infarction; NSTEMI, non-ST-segment elevation myocardial infarction; PCI, percutaneous coronary intervention; STEMI, ST-segment elevation myocardial infarction; TIA, transient ischaemic attack.

or history of revascularisation showed broadly similar results between subgroups (see online supplementary data, focusing on the CIs); however, precision was low due to the limited number of patients with a history of revascularisation or cancer.

## DISCUSSION

In this large population-based study using linked EHRs, 23.6% of patients who survived a first MI developed HF during a median follow-up of 3.7 years, resulting in an incidence rate of 64 cases per 1000 person-years. Incident HF was associated with increasing age, higher socioeconomic deprivation, a history of diabetes, atrial fibrillation, peripheral arterial disease, COPD, STEMI at presentation, BMI and smoking.

A previous Canadian study, using data from the period 1994–2000,[17] found that 71% of elderly patients without HF at index admission developed HF within 5 years' time after an MI, whereas the mortality due to MI decreased in the same period. The Framingham Heart Study,[18] using data from 1990 to 1999, found a 5-year post-MI HF incidence of 31.9% after MI; lower than found in the Canadian study. Importantly, both studies showed a higher incidence of post-MI HF than our more contemporary English cohort. This lower HF incidence is likely due to continued improvements of MI treatment strategies, which are reflected in a decreased HF incidence over calendar time. For example, a 20 812 sample of patients with MI hospitalised in Western Australia showed that the overall 1-year incidence of HF after MI decreased from 28.1% in 1996 to 16.5% in 2007.[19] This decline is confirmed further by a national Swedish hospital discharge and death registry study reporting a one-third decline in incidence between 1993 and 2004.[20] The same Swedish group recently reported that this trend persists for the period 2004–2013[2] and showed improved pharmacological treatment and early revascularisation in this period. During a median follow-up of 4 years, 19% of the patients were rehospitalised because of HF. An explanation for the higher percentage in our study is probably

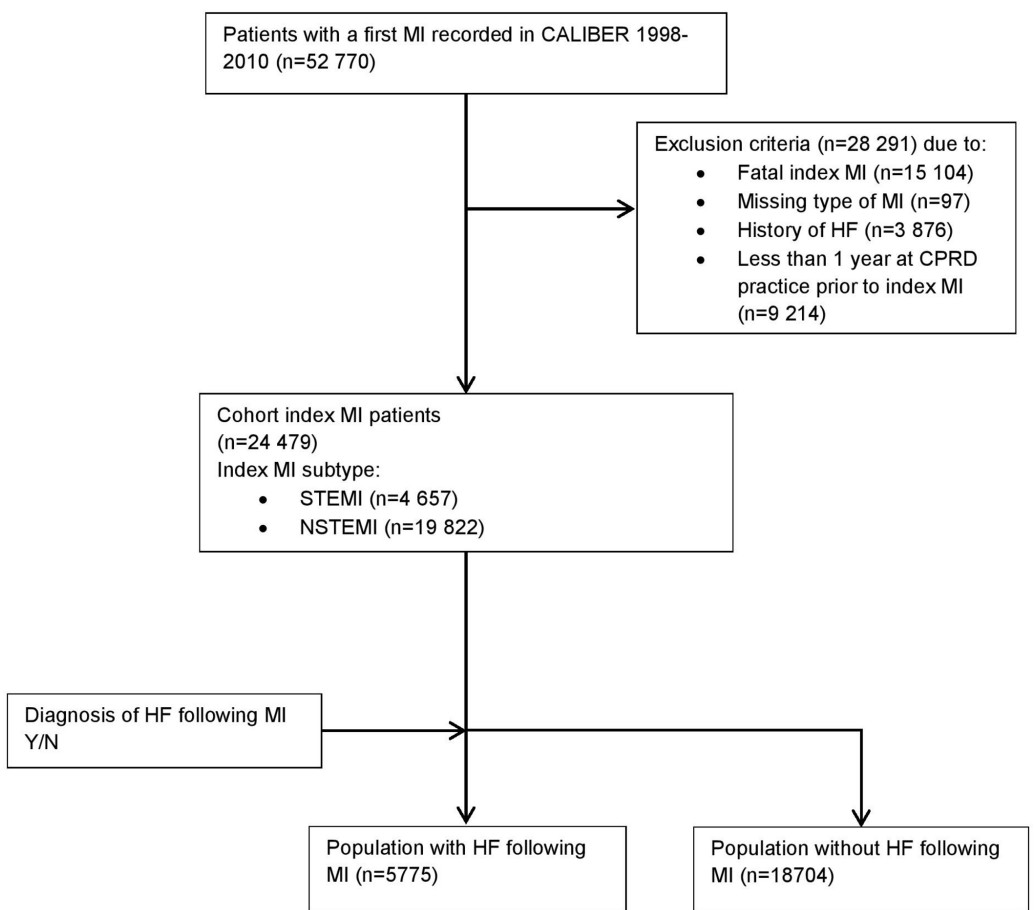

**Figure 1** Flow chart of study population CALIBER. Cardiovascular disease research using Linked Bespoke studies and electronic health records. CPRD, Clinical Practice Research Datalink; HF, heart failure; MI, myocardial infarction; NSTEMI, non-ST-segment elevation myocardial infarction; STEMI, ST-segment elevation myocardial infarction.

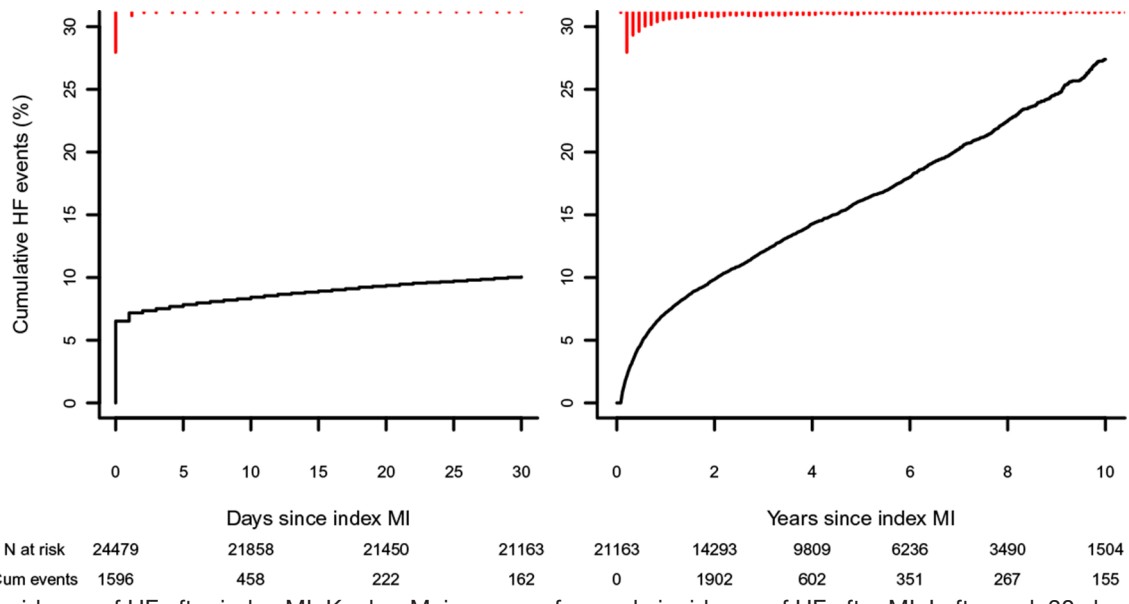

**Figure 2** Incidence of HF after index MI. Kaplan-Meier curves for crude incidence of HF after MI. Left panel: 30-day follow-up after index MI. Right panel: 10-year follow-up (median follow-up time was 3.7 years) in patients who survived the first 30 days and did not develop HF during the first 30 days (30 days event free). 'n at risk' represents the number of subjects at risk at a certain time point. 'Cum events' represents the cumulative number of events since the previous time points. A spike histogram is provided (in red) at the top of the graphs providing information on the number of events across time. HF, heart failure; MI, myocardial infarction.

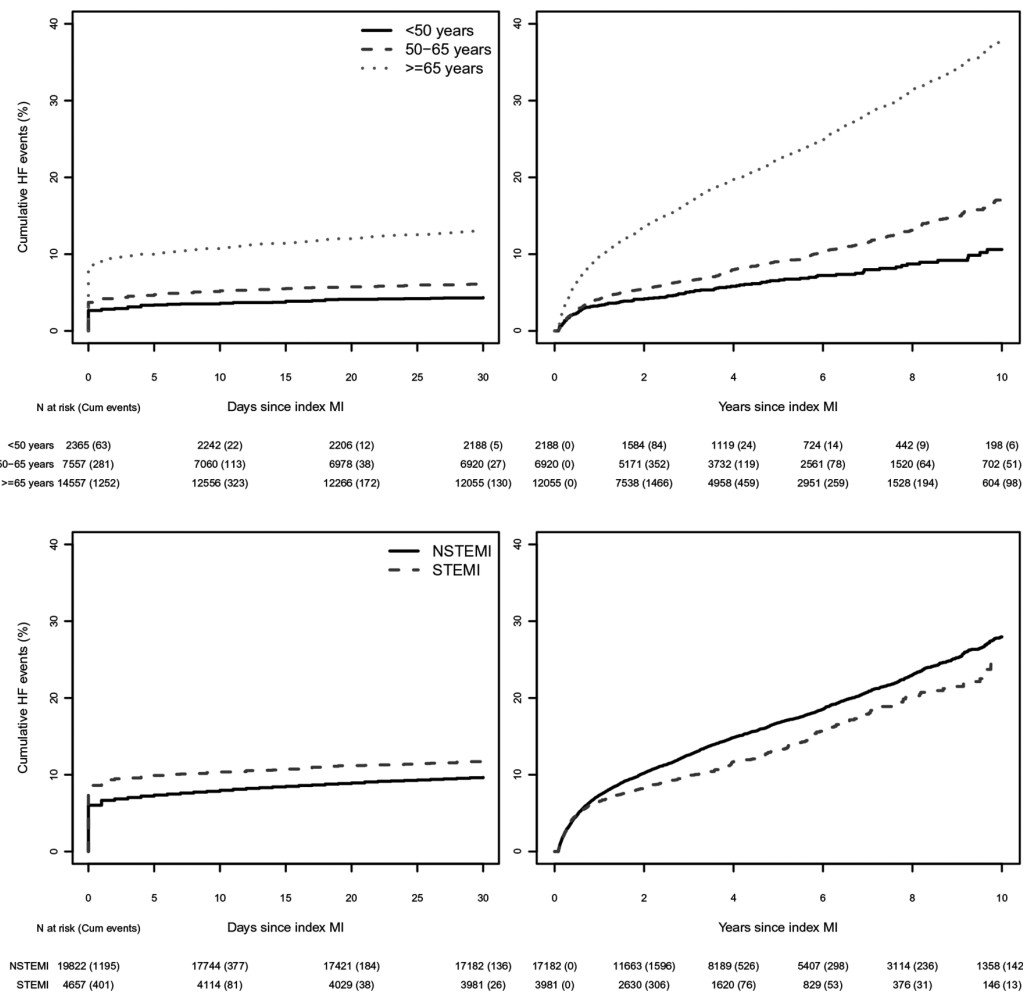

**Figure 3** Incidence of HF after index MI by group. Kaplan-Meier curves for crude incidence of HF after MI stratified by group. Top left panel: 30-day follow-up after index MI stratified by age group. Top right panel: 10-year follow-up (median follow-up time was 3.7 years) in 30-day event-free patients stratified by age group. Log-rank test of patients aged ≥65 years compared with patients aged 50–65 years (P<0.001). Bottom left panel: 30-day follow-up after index MI stratified by MI type. Bottom right panel: 10-year follow-up (median follow-up time was 3.7 years) in patients who survived the first 30 days and did not develop HF during the first 30 days stratified by MI type. Log-rank test of NSTEMI compared with STEMI (P<0.001). 'n at risk' represents the number of subjects at risk at a certain time point, with the cumulative number of events since the previous time points presented between brackets. HF, heart failure; MI, myocardial infarction; NSTEMI, non-ST-segment elevation myocardial infarction; STEMI, ST-segment elevation myocardial infarction.

## Prognostic factors

Similar to the current study, Torabi et al[21] reported that HF after MI increased steeply with age. Socioeconomic deprivation has been shown to be an independent predictor of HF development and associated with an increased incidence of HF in MI-free subjects.[22 23] Socioeconomic deprivation has also been associated with more frequent hospital admission and higher mortality in patients with HF.[24–26] The current study extends these observations to HF incidence after an MI. Furthermore, we showed that a history of diabetes, atrial fibrillation, peripheral arterial

the inclusion of HF diagnosis made in the GP setting next to hospital records decreasing misclassification of HF events. Future efforts are needed to harmonise these national data sources to compare daily care and HF epidemiology across different countries.

disease, COPD, STEMI at presentation, BMI and smoking are all independently prognostic of HF after MI. Interestingly, despite the size of the collected sample, sex only became significantly associated with HF after accounting for differences in BMI and smoking. Furthermore, the impact of male sex (HR 1.07, 95% CI 1.01 to 1.13) was modest indicating relative sex equality. It is of note that blood pressure was not significantly associated with HF. However, blood pressure measurements were frequently missing, which (even after multiple imputation) may be the cause of the observed lack of association, hence this deserves independent exploration. In this light, it is important to note that the diagnosis of hypertension was associated with HF. Potentially, these discrepancies between the association of blood pressure measurement and hypertension diagnosis can be explained by noting

**Table 2** HR for heart failure in patients following a first MI using multivariable Cox regression

| | Model 1 n=24 479 n events=5775 | Model 2 n=24 479 n events=5775 | Model 3 n=24 479 n events=5775 | Model 4 n=24 479 n events=5775 |
|---|---|---|---|---|
| | HR (95% CI) | HR (95% CI) | HR (95% CI) | HR (95% CI) |
| Age, per 10 years | 1.51 (1.48 to 1.55) | 1.49 (1.46 to 1.53) | 1.46 (1.43 to 1.50) | 1.45 (1.41 to 1.49) |
| Men | 1.06 (1.00 to 1.12) | 1.08 (1.02 to 1.14) | 1.07 (1.01 to 1.13) | 1.06 (1.00 to 1.12) |
| Index of Multiple Deprivation | | Overall  P≤0.001 | Overall P≤0.001 | Overall P≤0.001 |
| Q1 (least deprived) | | Reference | Reference | Reference |
| Q2 | | 1.09 (0.99 to 1.20) | 1.09 (0.99 to 1.20) | 1.08 (0.99 to 1.19) |
| Q3 | | 1.20 (1.09 to 1.33) | 1.20 (1.08 to 1.32) | 1.18 (1.07 to 1.31) |
| Q4 | | 1.20 (1.08 to 1.33) | 1.20 (1.08 to 1.33) | 1.17 (1.06 to 1.30) |
| Q5 (most deprived) | | 1.30 (1.16 to 1.45) | 1.30 (1.16 to 1.45) | 1.27 (1.13 to 1.41) |
| History of hypertension | | 1.19 (1.12 to 1.26) | 1.17 (1.10 to 1.24) | 1.16 (1.09 to 1.23) |
| History of diabetes | | 1.48 (1.38 to 1.60) | 1.49 (1.38 to 1.60) | 1.45 (1.35 to 1.56) |
| History of atrial fibrillation | | | 1.65 (1.53 to 1.77) | 1.62 (1.51 to 1.75) |
| Type of MI STEMI | | | 1.17 (1.10 to 1.26) | 1.19 (1.11 to 1.27) |
| History of peripheral arterial disease | | | | 1.38 (1.26 to 1.51) |
| History of COPD | | | | 1.28 (1.17 to 1.40) |
| Prescribed ACE inhibitor before MI | | | | 1.06 (0.99 to 1.13) |
| Prescribed angiotensin receptor blocker before MI | | | | 1.00 (0.89 to 1.11) |
| Prescribed beta-blocker before MI | | | | 0.93 (0.88 to 1.00) |

All analyses presented in table 2 were performed on a complete case dataset of 24 479 subjects.
COPD, chronic obstructive pulmonary disease; MI, myocardial infarction; Q, quintile; STEMI, ST-segment elevation myocardial infarction.

that a recorded diagnosis is indicative of long-term hypertensions, which may be different than a single blood pressure measurement in terms of prognosis. Further, given that both variables were included in the same model, the observed difference in association may suggest that conditional on hypertension, blood pressure itself is only modestly associated to HF, if at all.

### Strengths and limitations
The linkage of multiple EHR sources from primary and hospital care allowed for the collection of a representative sample,[27] which enabled us to explore the prognostic value of routinely collected data in primary care records and to detect non-hospitalised HF cases. The population of CPRD practices has been shown broadly representative of the UK population.[4 28] In total, 226 GP practices consented to data linkage with HES, MINAP and ONS (containing 3.9% of the population of England in 2006). A potential limitation is that the ascertainment of cardiovascular outcomes was not based on clinical criteria (eg, validated questionnaires and properly conducted physical examinations), practices of medical coding will have changed over time and there could be subgroups of patients with left ventricular dysfunction without clinical symptoms. Calendar-dependent changes over time were accounted for by using time-period stratified Cox models. We further wish to highlight the infrequent use of percutaneous coronary intervention in patients with STEMI, which was shown to be representative of the slower uptake in England.[29] Similarly, 32% of the patients used antiplatelet drugs at baseline which may be a reflection of non-MI CVD burden. The Kaplan-Meier plots for subtype of MI indicate a possible violation of the proportional hazard assumption for this variable. However, these plots represent the pairwise associations between MI subtype and time to HF and, as such, assume that MI subtype is independent of other prognostic factors, which is known to be false. The importance of conditioning on covariates is underlined by noting the Kaplan-Meier plots indicating a protective effect of STEMI versus NSTEMI, which was significantly reversed after correcting for covariates

(models 3–4). Similarly, instead of using Kaplan-Meier plots to assess the proportional hazard assumption of the *crude* associations, we used Schoenfeld residuals to explore this assumption for the *conditional* associations (multivariable model 4), which did not show any violations. Based on model 4, the absolute correlation between the Schoenfeld residuals and time was <0.10 (eg, for MI subtype, this was 0.06) indicating an absence of relevant interaction by time (eg, non-proportionality of hazard). Residual confounding due to medication use (or other missing/mis-specified variables) might be another potential source of bias; however, our intention was not to perform a causal analysis between drug prescriptions and time to HF. As such, it is interesting to note that despite the large sample size, we did not observe a significant association of ACE, angiotensin receptor blocker or beta-blocker prescribed prior to the indexing MI event and time to HF. Relatedly, we acknowledge that we did not assess all potential predictors of post-MI HF, for example, due to the high degree of missingness, we did not explore the prognostic potential of time to revascularisation.[30]

We adhered to CPRD recommendations to obtain up-to-standard baseline data by excluding patients with less than 1 year at CPRD practice prior to index MI. Previously, Lewis and colleagues showed that 3 months after registration with a new practice, most patient characteristics were updated correctly, which approximated 100% after 1 year of follow-up.[31] Assuming that duration of the CPRD follow-up is independent of the relations assessed here, excluding such patients should not hamper generalisability of results. Using the first MI recorded in the database without a prior history of HF might have introduced bias due to left truncation (eg, some subjects may already have experienced an MI before enrolment).[32] However, CALIBER holds longitudinal records from primary and hospital care, making it unlikely a large part of the patients were misclassified as MI free and the 1 year of follow-up prior to entry further decreases the likelihood of misclassifying patients.[31] While the current data are adequate to identify subjects with a first MI, the subclassification of patients with MI into STEMI and NSTEMI, despite recent improvements, is known to be error prone.[33] As such, results for MI type need replication using higher quality data, in perhaps purposely designed studies. Due to our interest in HF occurrence after a first MI, selection bias (eg, index event bias) was introduced.[34] This index event bias does not impact the descriptive or prognostic value of the association presented and is mostly relevant if one wants to develop an intervention based on the associations presented, which was not the aim of this study. Additionally, we note that we reduced the influence of selection bias by accounting for dependencies between predictors. An important caveat of electronic healthcare records is that these data are predominantly focused on recording diagnoses and prescriptions but not on their complement (ie, who is not diseased or who did not receive a drug). As such, we have assumed that subjects without a recorded drug prescription or diagnoses were unexposed or free of (that specific) disease. Provided that electronic registration is required for a patient to fill a prescription, we can be fairly confident that we did not miss many prescribed treatments. However, it is likely that some subjects were misclassified as free of disease while in fact they were not. We've attempted to minimise this misclassification by linking data across multiple healthcare settings and data sources (MINAP, HES and CPRD and ONS).

A final limitation is that there is a possible delay between primary and hospital care records. Previous research has shown that MI events tend to be recorded in primary care after the HES or MINAP record.[6] The lower 30-day HF incidence in patients with unclassified MI primarily derived from primary care could be partly explained because of a delay in coding. Therefore, we showed cumulative incidence rates in patients (alive and HF event free within the first 30 days) from 30 days after index MI to account for a delay in recording of MI in primary care. We were unable to differentiate between HF with preserved ejection fraction and HF with reduced ejection fraction as we had no access to detailed (echocardiographic) parameters to assess diastolic dysfunction. It is likely, however, that the majority of our patients with HF had developed systolic dysfunction after MI.

## CONCLUSION

In this large cohort study using linked EHRs in England from primary and hospital care, about one in four people developed HF within a median of 4 years after surviving a first MI. Increasing age, higher socioeconomic deprivation, a history of hypertension, diabetes, atrial fibrillation, peripheral arterial disease, COPD, smoking and STEMI at presentation were independent determinants of new onset HF following MI.

**Author affiliations**
[1]Farr Institute of Health Informatics Research, UCL Institute of Health Informatics, University College London, London, UK
[2]Department of Cardiology, Division Heart and Lungs, University Medical Center Utrecht, Utrecht, The Netherlands
[3]Institute of Cardiovascular Science, Faculty of Population Health Sciences, University College London, London, UK
[4]Groningen Research Institute of Pharmacy, University of Groningen, Groningen, The Netherlands
[5]Medical Research Council Bioinformatics Centre, Leeds Institute of Biomedical and Clinical Sciences, University of Leeds, Leeds, UK
[6]Medical Research Council Bioinformatics Centre, Leeds Institute for Cardiovascular and Metabolic Medicine, University of Leeds, Leeds, UK
[7]Julius Center for Health Sciences and Primary Care, University Medical Center Utrecht, Utrecht University, Utrecht, The Netherlands
[8]National Heart & Lung Institute, Royal Brompton & Harefield Hospitals, Imperial College London, London, UK
[9]Durrer Center for Cardiogenetic Research, ICIN-Netherlands Heart Institute, Utrecht, The Netherlands

**Contributors** JMIHG, AFS, LP, SK, MPR, SD, ADS, RSP, CPG, AWH, JGC, HH and FWA contributed to the idea and design of the study. JMIHG extracted and prepared the data for analysis. AFS and JMIHG performed the analysis. JMIHG drafted the manuscript, with revisions from AFS, LP, SK, MPR, SD, ADS, RSP, CPG, AWH, JGC, HH and FWA. FWA is guarantor

**Funding** This study was supported by the National Institute for Health Research (RP-PG-0407-10314), the Wellcome Trust (086091/Z/08/Z), the Medical Research Council Prognosis Research Strategy Partnership (G0902393/99558) and the Farr Institute of Health Informatics Research, funded by the Medical Research Council (MR/K006584/1), in partnership with the Arthritis Research UK, the British Heart Foundation, the Cancer Research UK, the Economic and Social Research Council, the Engineering and Physical Sciences Research Council, the National Institute of Health Research, the National Institute for Social Care and Health Research (Welsh Assembly Government), the Chief Scientist Office (Scottish Government Health Directorates) and the Wellcome Trust. Part of this work is funded through the Innovative Medicines Initiative 2 Joint Undertaking under grant agreement no 116074, BigData@Heart. AFS is funded by UCLH NIHR Biomedical Research Centre and is a UCL Springboard Population Health Sciences Fellow. HH is an NIHR Senior Investigator and receives support from the NIHR University College Hospitals/ University College London Biomedical Research Centre. FWA is supported by a Dekker scholarship-Junior Staff Member 2014T001-Netherlands Heart Foundation and UCL Hospitals NIHR Biomedical Research Centre.

**Competing interests** None declared.

**Patient consent** Obtained.

**Ethics approval** CALIBER has received ethics approval (supplementary methods). This study is in compliance with the Declaration of Helsinki, was approved by the ISAC (Independent Scientific Advisory Committee) for MHRA Database Research (protocol no 14_198R) and the MINAP Academic Group and is registered with ClinicalTrials.gov (NCT02384213).

**Provenance and peer review** Not commissioned; externally peer reviewed.

**Data sharing statement** Access to raw data can be requested from the CPRD (http://cprd.com).

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
