## [Reviewer comments · BMJ Open]

ARTICLE DETAILS

TITLE (PROVISIONAL)	An Electronic Health Records cohort study on Heart Failure Following Myocardial Infarction in England: Incidence and Predictors
AUTHORS	Gho, Johannes; Schmidt, Amand; Pasea, Laura; Koudstaal, Stefan; Pujades Rodriguez, Mar; Denaxas, S; Shah, Anoop; Patel, Riyaz; Gale, Chris; Hoes, Arno; Cleland, J; Hemingway, Harry; Asselbergs, Folkert

VERSION 1 – REVIEW

REVIEWER	Zhen Wang Mayo Clinic, USA
REVIEW RETURNED	01-Jul-2017

GENERAL COMMENTS	This study evaluated the incidence and prognostic factors of heart failure following MI. The statistical analyses were well conducted and presented. I have a few comments and suggestions. 1. Please add sentences on why the dataset is “representative”.2. Please add information about average sample size by calendar period and practice.3. Please describe why all figures were truncated at year 5. The dataset spans more than 10 years.4. Figures, please add events below number of patients at risk.5. Table 2, please confirm if n=24,479 for all models? There are no missings?6. Please describe what is “competing risk” and describe the findings for Appendix figure.7. Please describe how underweight/normal.. were defined.
---

REVIEWER	Daniel Sewell University of Iowa, USA
REVIEW RETURNED	14-Jul-2017

GENERAL COMMENTS	The authors have performed a thorough analysis of the data, implementing appropriate statistical procedures and imputation techniques, and verifying the modeling assumptions. They have also performed a sensitivity analysis to confirm the results. The analyses have been done completely and correctly, and I recommend accepting the article as is. The only comment I have is that the nested models seem redundant and unnecessary. Model 4 clearly contains some important variables; what then does this imply about the previous results from the first three models which omit these salient covariates?
--

	As the authors state at the top of p8, the focus is on the final model, so I would recommend reporting only model 4.
--	--

REVIEWER	Michelle D. Schmiegelow Department of Cardiology, Gentofte University Hospital, Copenhagen, Denmark
REVIEW RETURNED	17-Jul-2017

GENERAL COMMENTS	The authors of this observational, registry-based cohort study used the CALIBER database to assess the incidence of heart failure in 24,745 adult patients without prior MI or heart failure who survived first myocardial infarction (NSTEMI or STEMI) during January 1, 1998 to March 25, 2010. The CALIBER programme linked electronic health records for patients from 244 consenting general practices (primary care, Clinical Practice Research Datalink [CPRD]) to data from hospital administrative registers, Hospital Episode Statistics (HES), the mortality register from UK Office for National Statistics and the Myocardial Ischemia National Audit Project (MINAP). Patients were followed until death, March 25, 2010 or ceased registration at the participating GP. Over a median follow-up of 3.7 years, 23.6% (63.8/1,000 person-years) developed heart failure with atrial fibrillation, age, diabetes, peripheral arterial disease, greater socioeconomic deprivation, STEMI at presentation and hypertension being associated with increased risk, and underweight and overweight at baseline being associated with decreased risk of developing heart failure. The major strengths of the paper: The paper is well-written, and covers an important subject given the many advantages in the ischemia area primarily prompt revascularization, the comprehensive dataset used including clinical variables such as site of infarction, time to reperfusion and various clinical parameters such as LDL-cholesterol levels, and the choice of statistical method given the focus on prognostic predictors at time of MI presentation. Concerns of general character  1. I lack information regarding the population being studied which makes it difficult to interpret and extrapolate the results:  a. What were the selection criteria and characteristics of the consenting GP practices? 2. Given that the paper is registry-based, the validity of the results is determined by the validity of the data variables, particularly the definition and validity of the myocardial infarction diagnosis (identifying) and heart failure (the outcome variable). Please add information on the following:  a. Myocardial infarction: Does the index-myocardial infarction diagnosis derive from hospital records or GP registration? Which diagnosis codes were used? Has the diagnosis been validated? If it was a clinical diagnosis, which criteria was the diagnosis based on? The definition of MI changed in year 2000. Please comment. b. Heart failure: I cannot find any information on the definition of heart failure. A Danish study found that the sensitivity of the heart failure diagnosis in administrative health records is quite low (across multiple departments)[1]. In the discussion, the authors state the importance of using GP registrations for the diagnosis of heart failure suggesting that this was the source of outcome identification, but which criteria was the diagnosis based on? Do the authors have access to information on echocardiographic findings and/or NYHA class?
--

c. Diabetes: Did the diagnosis derive from GP registrations, and thus probably a clinical diagnosis based on HbA1c? – or was the diagnosis based on drug prescription, diagnosis codes, patient report, ...? How/when was the diagnosis included in the registry e.g. if the diagnosis was given at the patient's previous GP?

d. Hypertension: Was the diagnosis based on blood pressure measurements, prescribed and claimed antihypertensive medication or patient report?

e. Social deprivation: What was socioeconomic status based on (household income, highest attained education)? Please elaborate and consider using a more common terminology, since "higher deprivation" is a bit confusing.

3. Many of the clinical variables have a high percentage of missing values (>50%), whereas other baseline characteristics included in the models have 0% missing values, and I lack information about the discrepancy. Particularly since many of the variables with zero missing values are listed in the supplemental data as including imputed data, and the population used for the additional complete case analysis without imputed data (Model 5) only included 4,898 patients (20% of the study population). Please comment.

4. Please elaborate on the exclusion of patients with prior MI, and explore the importance of pre-existing ischemic heart disease. More specifically comment on the following:

a. A significant part of the population (15%) has had previous PCI or CABG, and thus must be assumed to have pre-existing ischemic heart disease, and maybe even prior MI.

b. One third (32%) of the population receives antiplatelet agents at baseline.

c. It is unclear if the exclusion of patients with prior MI only involves MIs in the year up to study inclusion.

5. Availability of early revascularization is the primary advantage in the handling of myocardial infarction patients, and it has been hypothesized that due to increased survival following MI, the incidence of heart failure may have increased, but conversely, that early revascularization decrease the risk of developing heart failure.[2-5]

6. In the context of the comment above, I lack important information about revascularization of the population.

a. Is the reporting of clinical variables regarding the index-MI (revascularization, time to reperfusion) mandatory or voluntary? Please add information on the procedure of registration.

b. Only one fifth of the STEMI-population received primary PCI. Is that representable of the UK?[6] Please also comment on the generalizability to other populations.

7. Although the authors address the crossing of NSTEMI/STEMI-curves, I find the potential violation of the hazard assumption in the multivariable Cox analysis concerning, particularly since the authors report STEMI-patients to have a higher short-term risk of heart failure (30-day), but a lower long-term risk of heart failure compared to NSTEMI-patients, but conclude that STEMI-patients have increased risk of developing heart failure? Please elaborate.

8. What does the study add? I have found several recent studies exploring risk of heart failure following acute myocardial infarction. Please add some of them and discuss. [4, 7-11]

Minor comments

9. Methods

a. Please comment on the risk of confounding by indication for the patients with data on biomarkers

b. Consider writing out EHRs

c. Patients with a history of cancer are included in the study, although the etiology of heart failure may be different in these patients. Please comment, and consider involving cancer in the multivariable analysis.

10. Results:

a. Line 171: please correct the number of patients from 24,701 to 24,479 patients

b. Lines 172-174: The percentages are misleading giving the high number of missing values. Please add the percentages of missing values to the brackets.

c. Line 178: Please add the missing space.

d. Change multiply to multiple.

e. Please add a plural "s" to factor.

11. Discussion:

a. Line 227: Regarding the missing association with hypertension, the authors refer to the missing values of blood pressure measurements. Conversely, it is reported that no patients had missing data on hypertension. The association between hypertension and risk of heart failure is well established and should be underscored particularly given the uncertainty of the findings. Please comment.

b. Line 242: The authors make a great effort arguing for the validity of the model despite the crossing curves. Nonetheless, I would like the authors to elaborate on the comment regarding that the Kaplan-Meier plots "assume that the predictors are unrelated to other prognostic factors, which is known to be false".

12. Table 2: Please clarify if imputed data was used in the models.

13. Figure 1: Please comment on the surprisingly large number of patients (9,214 patients, 17% of study population) excluded due to less than 1 year at CPRD practice prior to index MI.

14. Figure 2:

a. It would be interesting to see the 1-year absolute risk of heart failure.

b. More than 4,000 30-day survivors of MI are censored during the first year. What were the reasons for censoring?

15. Figure 3: The analyses stratified by STEMI/NSTEMI are important, but concerning given the crossing curves (please see comment 7). Please add an enlarged figure of the first year to clarify when the curves cross in order to allow for proper stratification.

REFERENCES

1. Kumler T, Gislason GH, Kirk V, Bay M, Nielsen OW, Kober L, Torp-Pedersen C: Accuracy of a heart failure diagnosis in administrative registers. *Eur J Heart Fail* 2008, 10(7):658-660.

2. Levy D, Kenchaiah S, Larson MG, Benjamin EJ, Kupka MJ, Ho KK, Murabito JM, Vasan RS: Long-term trends in the incidence of and survival with heart failure. *N Engl J Med* 2002, 347(18):1397-1402.

3. Ezekowitz JA, Kaul P, Bakal JA, Armstrong PW, Welsh RC, McAlister FA: Declining in-hospital mortality and increasing heart failure incidence in elderly patients with first myocardial infarction. *J Am Coll Cardiol* 2009, 53(1):13-20.

4. Velagaleti RS, Pencina MJ, Murabito JM, Wang TJ, Parikh NI, D'Agostino RB, Levy D, Kannel WB, Vasan RS: Long-term trends in the incidence of heart failure after myocardial infarction. *Circulation* 2008, 118(20):2057-2062.

	5. McAlister FA, Quan H, Fong A, Jin Y, Cujec B, Johnson D: Effect of invasive coronary revascularization in acute myocardial infarction on subsequent death rate and frequency of chronic heart failure. Am J Cardiol 2008, 102(1):1-5. 6. McNamara RL, Chung SC, Jernberg T, Holmes D, Roe M, Timmis A, James S, Deanfield J, Fonarow GC, Peterson ED et al: International comparisons of the management of patients with non-ST segment elevation acute myocardial infarction in the United Kingdom, Sweden, and the United States: The MINAP/NICOR, SWEDEHEART/RIKS-HIA, and ACTION Registry-GWTG/NCDR registries. Int J Cardiol 2014, 175(2):240-247. 7. Hung J, Teng TH, Finn J, Knuiman M, Briffa T, Stewart S, Sanfilippo FM, Ridout S, Hobbs M: Trends from 1996 to 2007 in incidence and mortality outcomes of heart failure after acute myocardial infarction: a population-based study of 20,812 patients with first acute myocardial infarction in Western Australia. Journal of the American Heart Association 2013, 2(5):e000172. 8. Sulo G, Igland J, Vollset SE, Nygard O, Ebbing M, Sulo E, Egeland GM, Tell GS: Heart Failure Complicating Acute Myocardial Infarction; Burden and Timing of Occurrence: A Nation-wide Analysis Including 86 771 Patients From the Cardiovascular Disease in Norway (CVDNOR) Project. Journal of the American Heart Association 2016, 5(1). 9. Shafazand M, Rosengren A, Lappas G, Swedberg K, Schaufelberger M: Decreasing trends in the incidence of heart failure after acute myocardial infarction from 1993-2004: a study of 175,216 patients with a first acute myocardial infarction in Sweden. Eur J Heart Fail 2011, 13(2):135-141. 10. Desta L, Jernberg T, Lofman I, Hofman-Bang C, Hagerman I, Spaak J, Persson H: Incidence, temporal trends, and prognostic impact of heart failure complicating acute myocardial infarction. The SWEDEHEART Registry (Swedish Web-System for Enhancement and Development of Evidence-Based Care in Heart Disease Evaluated According to Recommended Therapies): a study of 199,851 patients admitted with index acute myocardial infarctions, 1996 to 2008. JACC Heart Fail 2015, 3(3):234-242. 11. Desta L, Jernberg T, Spaak J, Hofman-Bang C, Persson H: Heart failure with normal ejection fraction is uncommon in acute myocardial infarction settings but associated with poor outcomes: a study of 91,360 patients admitted with index myocardial infarction between 1998 and 2010. Eur J Heart Fail 2016, 18(1):46-53.
--	--

VERSION 1 – AUTHOR RESPONSE

Reviewer 1

Reviewer Name: Zhen Wang

Institution and Country: Mayo Clinic, USA Please state any competing interests: None declared

Please leave your comments for the authors below

This study evaluated the incidence and prognostic factors of heart failure following MI. The statistical analyses were well conducted and presented. I have a few comments and suggestions.

1. Please add sentences on why the dataset is “representative”.

Response: The following explanation was added

Page 15 “The linkage of multiple electronic health record sources from primary and hospital care allowed for the collection of a representative sample[26], which enabled us to explore the prognostic value of routinely collected data in primary care records and to detect non-hospitalised HF cases. The population of CPRD practices has been shown broadly representative of the UK population[6,27]. In total 226 GP practices consented to data linkage with Hospital Episode Statistics, MINAP, and the Office for National Statistics (containing 3.9% of the population of England in 2006).”

2. Please add information about average sample size by calendar period and practice.

Response: Apologies for overlooking this admittedly important information.

Pages 9-10 “Patients were recruited from 226 GP practices, with the median practice enrolling 101 patients (quartile (Q1) 65 and Q3 150). Of the included subjects 4,657 were STEMI and 19,822 NSTEMI patients (0% missing), 15,969 (65.2%) were men (0% missing), 4,538 (42.1%) currently smoked (56% missing), 12,258 (50.1%) had hypertension (0% missing), and 3,014 (12.3%) had a history of diabetes at baseline (0% missing). In total, 6,129 (25.0%) were prescribed beta-blockers (0% missing) and 5,039 (20.6%) angiotensin converting enzyme (ACE) inhibitors (0% missing) prior to their indexing MI (Table 1). The number of patients identified across the previously defined calendar periods was reasonably stable, with a minimum of 5219 subjects in the period 1998-2001 compared to a maximum of 6838 subjects identified in the period 2004-2007.”

3. Please describe why all figures were truncated at year 5. The dataset spans more than 10 years.

Response: Our sincere gratitude for spotting this oversight. We have updated the figures to include the additional 5 years of follow-up. In the new figures, the curves are truncated after 10 years because of the small number of remaining patients. In the same light, we note the median follow-up of 3.7 years is considerably shorter than 10 years and we changed figure captions to highlight this.

4. Figures, please add events below number of patients at risk.

Response: The cumulative number of events (between two time points) was added to figures 2 and 3. Additionally, to further communicate the distribution of the number of events across time, we added a spiked histogram to the top of the graphs in figure 2.

5. Table 2, please confirm if n=24,479 for all models? There are no missings?

Response: The results presented in table 2 were indeed based on the 24,479 subjects, the following was added to the legend of table 2:

“All analyses presented in table 2 were performed on a complete case dataset of 24,479 subjects”

6. Please describe what is “competing risk” and describe the findings for Appendix figure.

Response: The following was added to describe the issue of competing risk

Page 9 "To account for the fact that patients may have died before the onset of HF (e.g., competing risk by mortality), analyses were repeated using Fine and Gray models[13,14]. As a hypothetical example of competing risk, let us assume that out of 100 MI subjects, one develops HF and 99 die. In a Cox's regression analysis all 99 dead subjects would be censored (implicitly, and incorrectly, assuming they may develop HF later in time), and the hazard of HF would equal 1/1. Instead a Fine and Gray model recognizes the 99 died subjects could never develop HF and hence calculates the hazard as 1/100."

Appendix figure is discussed on page 10 "During the first 30 days of follow-up only 1 patient died, limiting the potential for competing risk by all-cause mortality. In patients who did not have HF after 30 days accounting for competing risk by all-cause mortality attenuated the cumulative risk to be about 15% after 10 years of follow-up (Appendix Figure). Furthermore, the cumulative risks of all-cause mortality and HF converged as follow-up time progressed towards 10 years."

7. Please describe how underweight/normal.. were defined.

Response: The following was added to table 1 and appendix table 2.

Body mass index (BMI in kg/m²)
Underweight (18.5 <)
Normal (18.5; 25)
Overweight (25, 30)
Obese (>30)

Reviewer 2

Reviewer Name: Daniel Sewell

Institution and Country: University of Iowa, USA Please state any competing interests: None declared.

Please leave your comments for the authors below

The authors have performed a thorough analysis of the data, implementing appropriate statistical procedures and imputation techniques, and verifying the modeling assumptions. They have also performed a sensitivity analysis to confirm the results.

The analyses have been done completely and correctly, and I recommend accepting the article as is. The only comment I have is that the nested models seem redundant and unnecessary. Model 4 clearly contains some important variables; what then does this imply about the previous results from the first three models which omit these salient covariates? As the authors state at the top of p8, the focus is on the final model, so I would recommend reporting only model 4.

Response: We thank the reviewer for this suggestion and include an explanation of the relevance of models 1-3 relative to the "final" model 4.

Page 11 "Table 2 describes the results of the previously described nested Cox's models, showing little change in estimates between the increasingly complex models. Focusing on the final model, [...]"

Reviewer 3

Reviewer Name: Michelle D. Schmiegelow

Institution and Country: Department of Cardiology, Gentofte University Hospital, Copenhagen, Denmark Please state any competing interests: None declared

Please leave your comments for the authors below

The authors of this observational, registry-based cohort study used the CALIBER database to assess the incidence of heart failure in 24,745 adult patients without prior MI or heart failure who survived first myocardial infarction (NSTEMI or STEMI) during January 1, 1998 to March 25, 2010. The CALIBER programme linked electronic health records for patients from 244 consenting general practices (primary care, Clinical Practice Research Datalink [CPRD]) to data from hospital administrative registers, Hospital Episode Statistics (HES), the mortality register from UK Office for National Statistics and the Myocardial Ischemia National Audit Project (MINAP). Patients were followed until death, March 25, 2010 or ceased registration at the participating GP.

Over a median follow-up of 3.7 years, 23.6% (63.8/1,000 person-years) developed heart failure with atrial fibrillation, age, diabetes, peripheral arterial disease, greater socioeconomic deprivation, STEMI at presentation and hypertension being associated with increased risk, and underweight and overweight at baseline being associated with decreased risk of developing heart failure.

The major strengths of the paper:

The paper is well-written, and covers an important subject given the many advantages in the ischemia area primarily prompt revascularization, the comprehensive dataset used including clinical variables such as site of infarction, time to reperfusion and various clinical parameters such as LDL-cholesterol levels, and the choice of statistical method given the focus on prognostic predictors at time of MI presentation.

Concerns of general character

1. I lack information regarding the population being studied which makes it difficult to interpret and extrapolate the results:

a. What were the selection criteria and characteristics of the consenting GP practices?

Response: The following was added on page 15

“The linkage of multiple electronic health record sources from primary and hospital care allowed for the collection of a representative sample[26], which enabled us to explore the prognostic value of routinely collected data in primary care records and to detect non-hospitalised HF cases. The population of CPRD practices has been shown broadly representative of the UK population[6,27]. In total 226 GP practices consented to data linkage with Hospital Episode Statistics, MINAP, and the Office for National Statistics (containing 3.9% of the population of England in 2006).”

2. Given that the paper is registry-based, the validity of the results is determined by the validity of the data variables, particularly the definition and validity of the myocardial infarction diagnosis (identifying) and heart failure (the outcome variable). Please add information on the following:

a. Myocardial infarction: Does the index-myocardial infarction diagnosis derive from hospital records or GP registration? Which diagnosis codes were used? Has the diagnosis been validated? If it was a clinical diagnosis, which criteria was the diagnosis based on? The definition of MI changed in year 2000. Please comment.

Response: The following was added to the methods.

Page 7 “MI was defined by linking all of the CALIBER data sources (e.g., Hospital Episode Statistics and CPRD), with the type of MI recorded in MINAP and CPRD, see for more detail <https://www.caliberresearch.org> and the appendix. Characteristics regarding the index MI (e.g., type of revascularisation and delay from symptom onset to delivery of reperfusion therapy) were derived from MINAP. This nationwide registry is part of an annual audit which records over 20 key MI variables [9]. Herrett and colleagues [4] validated the current approach and showed the necessity of linking multiple sources to ascertain MI.”

We elaborated the discussion to address the potential inadequacies of this approach.

Page 16 “While the current data is adequate to identify subjects with a first MI, the sub-classification of MI patients into STEMI and NSTEMI, despite recent improvements, is known to be error prone [31]. As such, results for MI type need replication using higher quality data, in perhaps purposely designed studies.”

b. Heart failure: I cannot find any information on the definition of heart failure. A Danish study found that the sensitivity of the heart failure diagnosis in administrative health records is quite low (across multiple departments)[1]. In the discussion, the authors state the importance of using GP registrations for the diagnosis of heart failure suggesting that this was the source of outcome identification, but which criteria was the diagnosis based on? Do the authors have access to information on echocardiographic findings and/or NYHA class?

Response: Our apologies for this important omission, the following was included

Page 7 “The primary study outcome was the first HF event following MI, which similar to MI, was recorded in multiple CALIBER data sources; echocardiographic findings and NYHA class were unavailable. All phenotypes were created using a robust methodology described elsewhere [10].”

c. Diabetes: Did the diagnosis derive from GP registrations, and thus probably a clinical diagnosis based on HbA1c? – or was the diagnosis based on drug prescription, diagnosis codes, patient report,...? How/when was the diagnosis included in the registry e.g. if the diagnosis was given at the patient’s previous GP?

Response: Page 7 “Diabetes and hypertension diagnosis were based on Read codes from primary care or ICD10 codes from Hospital Episode Statistics, both of which were classified using primary care consultation records and hospitals diagnosis records. “

d. Hypertension: Was the diagnosis based on blood pressure measurements, prescribed and claimed antihypertensive medication or patient report?

Response: See our response to comment 2c

e. Social deprivation: What was socioeconomic status based on (household income, highest attained education)? Please elaborate and consider using a more common terminology, since “higher deprivation” is a bit confusing.

Response: Page 7 “Socioeconomic status was based on the Index of Multiple Deprivation which includes seven domains of deprivation (income deprivation, employment deprivation, health deprivation and disability, education, skills and training deprivation, barriers to housing and services, living environment deprivation and crime) [8].”

3. Many of the clinical variables have a high percentage of missing values (>50%), whereas other baseline characteristics included in the models have 0% missing values, and I lack information about the discrepancy. Particularly since many of the variables with zero missing values are listed in the supplemental data as including imputed data, and the population used for the additional complete case analysis without imputed data (Model 5) only included 4,898 patients (20% of the study population). Please comment.

Response: Model 5 was implemented both as a complete case analyses (without multiple imputed data) with 4,898 subjects and additionally based on imputed data with the same number of 24,479 subject used in the other analyses. Reassuringly there was limited difference in point estimates between the two versions of model 5.

Page 11 “After multiply imputing the data models were extended to include smoking, BMI, and systolic and diastolic blood pressure variables which showed BMI, male gender and smoking to be conditionally independent prognostic factors (Supplementary Table S2). The extended model 5 was implemented using imputed data, and non-imputed (complete case) data, resulting in similar HR estimates (magnitude and direction).”

The following was added to explain the difference in missing pattern between (biomarker) measurements and health record related variables

Page 8 “Due to the availability of electronic health care records from multiple sources, clinical diagnoses and prescriptions were completely recorded; electronic registration is required for a patient to fill a prescription. Biomarker and lifestyle measurements such as smoking, BMI, and blood pressure were however, incompletely recorded (see Table 1). It is probable that these data were preferentially recorded in subjects perceived to be at a higher risk for early progression of CVD. Pairwise analyses regressing a complete case indicator on observed variables indeed showed considerable dependency between recorded data and missingness (results not shown), violating the “missing completely at random” assumption. This dependency was utilized by multiply imputing missing data using the mice package, which was implemented using 20 imputed datasets and pooled based on Rubin’s rules (Supplementary)[12].”

As typical with imputing missing data, these were imputed on a mixture of fully observed and partially observed variables as explained in the appendix. The rationale being that even partially observed variables will improve the accuracy of the imputation models.

4. Please elaborate on the exclusion of patients with prior MI, and explore the importance of pre-existing ischemic heart disease. More specifically comment on the following:
a. A significant part of the population (15%) has had previous PCI or CABG, and thus must be assumed to have pre-existing ischemic heart disease, and maybe even prior MI.

Response: The potential impact of misclassification due to left truncation is discussed on page 16

“We adhered to CPRD recommendations to obtain up-to-standard baseline data by excluding patients with less than 1 year at CPRD practice prior to index MI. Previously Lewis and colleagues showed that 3 months after registration with a new practice most patient characteristics were updated correctly, which approximated 100% after 1 year of follow-up[29]. Assuming that duration of the CPRD follow-up is independent of the relations assessed here, excluding such patients should not hamper generalizability of results. Using the first MI recorded in the database without a prior history of HF, might have introduced bias due to left truncation (e.g., some subjects may already have

experienced an MI before enrolment)[30]. However CALIBER holds longitudinal records from primary and hospital care, making it unlikely a large part of the patients were misclassified as MI free and the one year of follow-up prior to entry further decreases the likelihood of misclassifying patients[29].”

b. One third (32%) of the population receives antiplatelet agents at baseline.

Response: The following was added to the discussion on page 15 “Similarly, 32% of the patients used antiplatelet drugs at baseline which may be a reflection of non-MI CVD burden“

c. It is unclear if the exclusion of patients with prior MI only involves MIs in the year up to study inclusion.

Response: Patients were follow-up from their first MI to a diagnosis of HF or lost to follow-up. Given that MI diagnoses were based on electronic health care records, a small minority of patients may have experienced a previous MI that was not registered. For changes made we refer to our response to comment 4c

5. Availability of early revascularization is the primary advantage in the handling of myocardial infarction patients, and it has been hypothesized that due to increased survival following MI, the incidence of heart failure may have increased, but conversely, that early revascularization decrease the risk of developing heart failure.[2-5]

Response: We agree that the incidence of HF following MI could be mainly influenced by two competing trends. Early revascularization therapy and increased uptake of drugs like ACE inhibitors and beta blockade have led to smaller infarct sizes and decreasing the risk of heart failure. This is recently demonstrated in the national Swedish registry (Desta L. Int J Cardiol. 2017 May 24. pii: S0167-5273(16)34867-7).

However, as shown in the paper you referenced by Ezekowith, the opposite is seen in elderly. The authors concluded that improved survival following MI was accompanied by an increase in HF incidence. In the current manuscript we did not focus on temporal trends, which is a topic of a follow-up study in perpetration, which will also look explore the timing of revascularization and risk of developing HF in detail.

We added the following paragraph to the discussion “A previous Canadian study, using data from the period 1994-2000[16], found that 71% of elderly patients without HF at index admission developed HF within 5 years’ time after an MI, whereas the mortality due to MI decreased in the same period. The Framingham Heart Study[17], using data from 1990 to 1999, found a 5-years post-MI HF incidence of 31.9% after MI; lower than found in the Canadian study. Importantly, both studies showed a higher incidence of post-MI HF than our more contemporary English cohort. This lower HF incidence is likely due to continued improvements of MI treatment strategies, which are reflected in a decreased HF incidence over calendar time.

For example, a 20,812 sample of MI patients hospitalized in Western Australia showed that the overall 1-year incidence of HF after MI decreased from 28.1% 1996 in to 16.5% in 2007[18]. This decline is confirmed further by a national Swedish hospital discharge and death registry study reporting a one third decline in incidence between 1993-2004[19].The same Swedish group recently reported that this trend persists for the period 2004-2013[2] and showed improved pharmacological treatment and early revascularisation in this period. During a median follow-up of 4 years, 19% of the patients were re-hospitalized because of HF. An explanation for the higher percentage in our study is probably the inclusion of HF diagnosis made in the GP setting next to hospital records decreasing

misclassification of HF events. Future efforts are needed to harmonize these national data sources to compare daily care and HF epidemiology across different countries.’

6. In the context of the comment above, I lack important information about revascularization of the population.

a. Is the reporting of clinical variables regarding the index-MI (revascularization, time to reperfusion) mandatory or voluntary? Please add information on the procedure of registration.

Response: This additional information is describe on page 7

“MI was defined by linking all of the CALIBER data sources (e.g., Hospital Episode Statistics and CPRD), with the type of MI recorded in MINAP and CPRD, see for more detail <https://www.caliberresearch.org> and the appendix. Characteristics regarding the index MI (e.g., type of revascularisation and delay from symptom onset to delivery of reperfusion therapy) were derived from MINAP. This nationwide registry is part of an annual audit which records over 20 key MI variables [9].”

b. Only one fifth of the STEMI-population received primary PCI. Is that representable of the UK?[6] Please also comment on the generalizability to other populations.

Response: Thank you; we elaborated the discussion to address this

Page 15 “We further wish to highlight the infrequent use of PCI in STEMI patients, which was shown to be representative of the slower uptake in England[28].”

7. Although the authors address the crossing of NSTEMI/STEMI-curves, I find the potential violation of the hazard assumption in the multivariable Cox analysis concerning, particularly since the authors report STEMI-patients to have a higher short-term risk of heart failure (30-day), but a lower long-term risk of heart failure compared to NSTEMI-patients, but conclude that STEMI-patients have increased risk of developing heart failure? Please elaborate.

Response: The following was added on pages 15-16

“The Kaplan-Meier plots for subtype of MI indicate a possible violation of the proportional hazard assumption for this variable. However, these plots represent the pairwise associations between MI subtype and time to HF, and as such assume that MI subtype is independent of other prognostic factors, which is known to be false. The importance of conditioning on covariates is underlined by noting the Kaplan-Meier plots indicating a protective effect of STEMI versus NSTEMI, which was significantly reversed after correcting for covariates (models 3-4). Similarly, instead of using Kaplan-Meier plots to assess the proportional hazard assumption of the crude associations, we used Schoenfeld residuals to explore this assumption for the conditional associations (multivariable model 4); which did not show any violations. Based on model 4 the absolute correlation between the Schoenfeld residuals and time was < 0.10 (e.g., for MI subtype this was 0.06) indicating an absence of relevant interaction by time (e.g., non-proportionality of hazard).”

8. What does the study add? I have found several recent studies exploring risk of heart failure following acute myocardial infarction. Please add some of them and discuss. [4, 7-11]

Response: Previous studies, as the ones referred to by the reviewer, exclusively focus on hospital records data. The major strength of our contribution is the combination of hospital and GP data. As such, the fact that previous studies using a very specific subset of patients found similar results should be seen as reassuring, with our study confirming that these findings can be generalized beyond hospital patients (which would remain unknown without the current study).

For changes made please see our reply to comment 5.

9. Methods

a. Please comment on the risk of confounding by indication for the patients with data on biomarkers

Response: The following was added regarding subset of patients with measured biomarkers.

Page 8 “ It is probable that these data were preferentially recorded in subjects perceived to be at a higher risk for early progression of CVD. Pairwise analyses regressing a complete case indicator on observed variables indeed showed considerable dependency between recorded data and missingness (results not shown), violating the “missing completely at random” assumption. This dependency was utilized by multiply imputing missing data using the mice package, which was implemented using 20 imputed datasets and pooled based on Rubin’s rules (Supplementary)[12].”

Confounding is discussed on page 16 “Residual confounding due to medication use (or other missing/mis-specified variables) might be another potential source of bias, however our intention was not to perform a causal analysis between drug prescriptions and time to HF. “

b. Consider writing out EHRs

Response: Done, thank you

c. Patients with a history of cancer are included in the study, although the etiology of heart failure may be different in these patients. Please comment, and consider involving cancer in the multivariable analysis.

Response: We wish to thank the reviewer for this suggestion and have included cancer-stratified models to explore the differences between prognostic factors between subjects with/without a history of cancer.

Page 15 “In addition to these sensitivity analyses, we performed a cancer-stratified analysis. Due to discrepancies in sample size between the two subgroups we decided against formal interaction testing, which may suffer from inflated false positive rates and lower power in such settings[15]. “

Page 11 “Stratifying the sample on cancer diagnosis status showed broadly similar results (See appendix table S3, focusing on the confidence intervals) between subject with and without a history of cancer. “

10. Results:

a. Line 171: please correct the number of patients from 24,701 to 24,479 patients

Response: Done with our thanks for spotting this mistake.

b. Lines 172-174: The percentages are misleading giving the high number of missing values. Please add the percentages of missing values to the brackets.

Response: We agree, this was implemented as suggested.

c. Line 178: Please add the missing space.

Response: Thank you for spotting this type-o

d. Change multiply to multiple.

Response: Thank you.

e. Please add a plural "s" to factor.

Response: Changed, with our thanks.

11. Discussion:

a. Line 227: Regarding the missing association with hypertension, the authors refer to the missing values of blood pressure measurements. Conversely, it is reported that no patients had missing data on hypertension. The association between hypertension and risk of heart failure is well established and should be underscored particularly given the uncertainty of the findings. Please comment.

Response: Thank you we agree, the following was changed

Page 13 "However, blood pressure measurements were frequently missing, which (even after multiple imputation) may be the cause of the observed lack of association hence this deserves independent exploration. In this light it is important to note that the diagnosis of hypertension was associated with HF. Potentially this discrepancies between the association of blood pressure measurement and hypertension diagnosis can be explained by noting that a recorded diagnosis is indicative of long-term hypertensions, which may be prognostically different than a single blood pressure measurement. Further, given that both variables were included in the same model, the observed difference in association may suggest that conditional on hypertension, blood pressure itself is only modestly associated to HF, if at all.

"

b. Line 242: The authors make a great effort arguing for the validity of the model despite the crossing curves. Nonetheless, I would like the authors to elaborate on the comment regarding that the Kaplan-Meier plots "assume that the predictors are unrelated to other prognostic factors, which is known to be false".

Response: We appreciate some readers may accustomed to judge proportional hazard using KM plots. However, this is generally an incorrect approach (unless analysing an RCT) and a better alternative is to plot the Schoenfeld residuals against time to look for trends, and perhaps additionally tests for use a correlation between these residuals (which reflect the time specific HR) and time. The benefit of using Schoenfeld residuals compared to KM plots is the former allows for conditioning on covariates while the later does not. In essence KM plots test if the PH assumption holds for the crude association, while we are interested in the validity of the PH assumption for the multivariable (conditional) models/associations.

For changes made to the manuscript please see our response to your comment 7 above.

12. Table 2: Please clarify if imputed data was used in the models.

Response: No imputed data were used for table 2. The following was added

"All analyses presented in table 2 were performed on a complete case dataset of 24,479 subjects."

13. Figure 1: Please comment on the surprisingly large number of patients (9,214 patients, 17% of study population) excluded due to less than 1 year at CPRD practice prior to index MI.

Response: This was discussed on page 16

“We adhered to CPRD recommendations to obtain up-to-standard baseline data by excluding patients with less than 1 year at CPRD practice prior to index MI. Previously Lewis and colleagues showed that 3 months after registration with a new practice most patient characteristics were updated correctly, which approximated 100% after 1 year of follow-up[29].”

14. Figure 2:

a. It would be interesting to see the 1-year absolute risk of heart failure.

Response: The following was added

Page 10 “From day 30 onwards 3,337 (15.8%) of MI patients (event free during the first 30 days) developed HF, with 6.8% experiencing an HF event within the first year”

b. More than 4,000 30-day survivors of MI are censored during the first year. What were the reasons for censoring?

Response: 2,635 subject were censored within 1 year out of 21163 subject at risk (surviving the first 30 days). This perhaps high number is simply a reflection of the ongoing recruitment of the CPRD and the fact that we are only starting follow-up after a first MI occurred (instead of registering with a GP practice). Only a minority of this censoring is due to actual fatalities (as shown in the appendix figure).

The figures (including the appendix figure) have been updated to present not only the number of subjects at risk but also the cumulative number of events (including mortality for the appendix) between two time points.

15. Figure 3: The analyses stratified by STEMI/NSTEMI are important, but concerning given the crossing curves (please see comment 7). Please add an enlarged figure of the first year to clarify when the curves cross in order to allow for proper stratification.

Response: See our answer to comment 7. Regarding the moment the two curves cross the following was added on pages 10.

“Excluding patients who experienced a HF within the first 30 days showed that STEMI patients had a lower incidence of HF than NSTEMI subjects (Figure 3 lower panels). At 57 days the crude cumulative risk of HF in the NSTEMI subjects surpassed that of the STEMI subjects for the first time (0.0151 versus 0.0147), with the curves further diverging at 73 days of follow-up. ”

REFERENCES

1. Kumler T, Gislason GH, Kirk V, Bay M, Nielsen OW, Kober L, Torp-Pedersen C: Accuracy of a heart failure diagnosis in administrative registers. *Eur J Heart Fail* 2008, 10(7):658-660.
2. Levy D, Kenchaiah S, Larson MG, Benjamin EJ, Kupka MJ, Ho KK, Murabito JM, Vasan RS: Long-term trends in the incidence of and survival with heart failure. *N Engl J Med* 2002, 347(18):1397-1402.

3. Ezekowitz JA, Kaul P, Bakal JA, Armstrong PW, Welsh RC, McAlister FA: Declining in-hospital mortality and increasing heart failure incidence in elderly patients with first myocardial infarction. *J Am Coll Cardiol* 2009, 53(1):13-20.
4. Velagaleti RS, Pencina MJ, Murabito JM, Wang TJ, Parikh NI, D'Agostino RB, Levy D, Kannel WB, Vasan RS: Long-term trends in the incidence of heart failure after myocardial infarction. *Circulation* 2008, 118(20):2057-2062.
5. McAlister FA, Quan H, Fong A, Jin Y, Cujec B, Johnson D: Effect of invasive coronary revascularization in acute myocardial infarction on subsequent death rate and frequency of chronic heart failure. *Am J Cardiol* 2008, 102(1):1-5.
6. McNamara RL, Chung SC, Jernberg T, Holmes D, Roe M, Timmis A, James S, Deanfield J, Fonarow GC, Peterson ED et al: International comparisons of the management of patients with non-ST segment elevation acute myocardial infarction in the United Kingdom, Sweden, and the United States: The MINAP/NICOR, SWEDEHEART/RIKS-HIA, and ACTION Registry-GWTG/NCDR registries. *Int J Cardiol* 2014, 175(2):240-247.
7. Hung J, Teng TH, Finn J, Knuiman M, Briffa T, Stewart S, Sanfilippo FM, Ridout S, Hobbs M: Trends from 1996 to 2007 in incidence and mortality outcomes of heart failure after acute myocardial infarction: a population-based study of 20,812 patients with first acute myocardial infarction in Western Australia. *Journal of the American Heart Association* 2013, 2(5):e000172.
8. Sulo G, Iglund J, Vollset SE, Nygard O, Ebbing M, Sulo E, Egeland GM, Tell GS: Heart Failure Complicating Acute Myocardial Infarction; Burden and Timing of Occurrence: A Nation-wide Analysis Including 86 771 Patients From the Cardiovascular Disease in Norway (CVDNOR) Project. *Journal of the American Heart Association* 2016, 5(1).
9. Shafazand M, Rosengren A, Lappas G, Swedberg K, Schaufelberger M: Decreasing trends in the incidence of heart failure after acute myocardial infarction from 1993-2004: a study of 175,216 patients with a first acute myocardial infarction in Sweden. *Eur J Heart Fail* 2011, 13(2):135-141.
10. Desta L, Jernberg T, Lofman I, Hofman-Bang C, Hagerman I, Spaak J, Persson H: Incidence, temporal trends, and prognostic impact of heart failure complicating acute myocardial infarction. The SWEDEHEART Registry (Swedish Web-System for Enhancement and Development of Evidence-Based Care in Heart Disease Evaluated According to Recommended Therapies): a study of 199,851 patients admitted with index acute myocardial infarctions, 1996 to 2008. *JACC Heart Fail* 2015, 3(3):234-242.
11. Desta L, Jernberg T, Spaak J, Hofman-Bang C, Persson H: Heart failure with normal ejection fraction is uncommon in acute myocardial infarction settings but associated with poor outcomes: a study of 91,360 patients admitted with index myocardial infarction between 1998 and 2010. *Eur J Heart Fail* 2016, 18(1):46-53.

VERSION 2 – REVIEW

REVIEWER	Zhen Wang Mayo Clinic, USA
REVIEW RETURNED	08-Sep-2017

GENERAL COMMENTS	The authors have addressed all of my comments. I have no further suggestions.
---

REVIEWER	Daniel Sewell University of Iowa
REVIEW RETURNED	05-Sep-2017

GENERAL COMMENTS	NA
----

REVIEWER	Michelle Dalgas Skøtt Schmiegelow Department of Cardiology Gentofte University Hospital Herlev-Gentofte Hospital Kildegårds Vej 28 2900 Hellerup Danmark
REVIEW RETURNED	09-Sep-2017

GENERAL COMMENTS	The authors have improved the paper significantly, but I still have some concerns, primarily the sparse information about use of revascularization, which is crucial for the interpretation and generalizability of the results (see “Discussion” below). Introduction:  • Lines 121–123: Please add that changes could also be explained by changes in definition of heart failure, an important factor in registry-based studies. Methods:  • Line 227: Please explain what prodcodes is? • Line 282: It is greatly appreciated that the authors have added information on validity. Please elaborate on “the necessity of multiple sources” and the result of combining data sources, i.e. sensitivity, specificity and positive predictive value. Further, please make a reference to the supplemental material elaborating on the used codes. • Line 285 (sensitivity analyses): I do not agree that the use of administrative registries ascertain that all clinical diagnoses are registered. Did the authors use only primary or also secondary diagnoses? Risk of misregistration must be added to the limitations. If this is not an issue in the UK system, this must be argued for. Results:  • Line 408: Given that the risk of heart failure according to type of MI changes over time from MI, the hazard ratios of heart failure by type of MI must be reported according to time since MI. • The authors argue that the lower risk of heart failure compared to studies from the 1990ies can be explained by improvements in MI treatment strategies. However, only characteristics about revascularization for STEMI patients are reported, and information about revascularization is crucial for the interpretation of the findings and generalizability. Discussion:  • Although the results according to type of MI must be interpreted with caution, and data on delay from symptoms to reperfusion have a large number of missing, it would be interesting to see characteristics according to type of MI. • The authors discuss the risk of bias due to left truncation. The authors should however test for interaction according to prior revascularization (15% of the MI population), as these individuals must have ischemic heart disease and thus a higher baseline risk of heart failure, or maybe even undiagnosed, pre-existing heart failure. Conclusion  • Given the large number of missing data for BMI, please omit conclusions about associations with weight. Further  • Please change “gender” to “sex” throughout the paper, i.e. (Oxford
--

	dictionary) Grammar (in languages such as Latin, French, and German) each of the classes (typically masculine, feminine, common, neuter) of nouns and pronouns distinguished by the different inflections which they have and which they require in words syntactically associated with them. Grammatical gender is only very loosely associated with natural distinctions of sex.
--	--

VERSION 2 – AUTHOR RESPONSE

Reviewer: 1

Reviewer Name: Zhen Wang

Institution and Country: Mayo Clinic, USA

Please state any competing interests: None declared

Please leave your comments for the authors below

The authors have addressed all of my comments. I have no further suggestions.

Response: Thank you

Reviewer: 2

Reviewer Name: Daniel Sewell

Institution and Country: University of Iowa

Please state any competing interests: None declared

Please leave your comments for the authors below

NA

Response: Thank you

Reviewer: 3

Reviewer Name: Michelle Dalgas Skøtt Schmiegelow

Institution and Country: Department of Cardiology, Gentofte University Hospital, Herlev-Gentofte Hospital

Kildegårds Vej 28, 2900 Hellerup, Danmark

Please state any competing interests: None declared

Please leave your comments for the authors below

The authors have improved the paper significantly, but I still have some concerns, primarily the sparse information about use of revascularization, which is crucial for the interpretation and generalizability of the results (see “Discussion” below).

Introduction:

- Lines 121–123: Please add that changes could also be explained by changes in definition of heart failure, an important factor in registry-based studies.

Response: Done, thank you for this suggestion.

Methods:

- Line 227: Please explain what prodcodes is?

Response: This section was rewritten to

Page 7 “Study variables were derived from diagnoses recorded across several controlled clinical terminology and statistical classification systems: Read, ICD-9 or ICD-10, medication prescription information or the Office of Population Censuses and Surveys Classification of Interventions and Procedures (OPCS-4) codes (see <https://rdr.io/rforge/CALIBERcodelists/man/PRODDICT.html>) with the electronic health record phenotyping algorithms published at <https://caliberresearch.org/portal> [5].“

- Line 282: It is greatly appreciated that the authors have added information on validity. Please elaborate on “the necessity of multiple sources” and the result of combining data sources, i.e. sensitivity, specificity and positive predictive value. Further, please make a reference to the supplemental material elaborating on the used codes.

Response: Reference 4: <http://www.bmj.com/content/bmj/346/bmj.f2350.full.pdf> for example, figure 3 depicts the overlap (or lack thereof) of non-fatal MI diagnoses in 3 databases (CPRD, MINAP, and HES), showing that only 31% in documented in all 3, hence arguing that multiple data sources are necessary.

As requested we've included a reference to the supplementary codes.

- Line 285 (sensitivity analyses): I do not agree that the use of administrative registries ascertain that all clinical diagnoses are registered. Did the authors use only primary or also secondary diagnoses? Risk of misregistration must be added to the limitations. If this is not an issue in the UK system, this must be argued for.

Response: We fully agree with this and as suggested include this important caveat in the limitations.

Page 15 “An important caveat of electronic health care records is that these data are predominantly focussed on recording diagnoses and prescriptions but not on their complement (i.e., who is not diseased or who did not receive a drug).

As such we have assumed that subjects without a recorded drug prescription or diagnoses was unexposed or free of (that specific) disease. Provided that electronic registration is required for a patient to fill a prescription we can be fairly confident that we did not miss many prescribed treatments. However, it is likely that some subjects were misclassified as free of disease while in fact they were not. We've attempted to minimize this misclassification by linking data across multiple healthcare settings and data sources (MINAP, HES and CPRD, ONS). “

Results:

- Line 408: Given that the risk of heart failure according to type of MI changes over time from MI, the hazard ratios of heart failure by type of MI must be reported according to time since MI.

Response: We've accounting for potential changes overtime by utilizing time-stratified Cox's models. Admittedly the time period used were relatively crude, however testing for interaction by time did not show any relevant deviations from additivity of effects over time.

The stratification is described on page 9:

“All models were stratified on general practice and calendar year periods of enrolment (1998-2001, 2001-2004, 2004-2007 and 2007-2010). Stratified models were used instead of frailty models because the former does not make any distributional assumption.”

The exploration of interactions by time is described on page 9:

“The proportional hazards assumption was checked using Schoenfeld residuals, and by nonparametric correlation coefficients between survival time and the parameter specific residuals”

Page 5

“ Similarly, instead of using Kaplan-Meier plots to assess the proportional hazard assumption of the crude associations, we used Schoenfeld residuals to explore this assumption for the conditional associations (multivariable model 4); which did not show any violations. Based on model 4 the absolute correlation between the Schoenfeld residuals and time was < 0.10 (e.g., for MI subtype this was 0.06) indicating an absence of relevant interaction by time (e.g., non-proportionality of hazard). “

- The authors argue that the lower risk of heart failure compared to studies from the 1990ies can be explained by improvements in MI treatment strategies. However, only characteristics about revascularization for STEMI patients are reported, and information about revascularization is crucial for the interpretation of the findings and generalizability.

Discussion:

Response: This was added

- Although the results according to type of MI must be interpreted with caution, and data on delay from symptoms to reperfusion have a large number of missing, it would be interesting to see characteristics according to type of MI.

Response: Patient characteristics according to the type of MI are presented in Table 1.

- The authors discuss the risk of bias due to left truncation. The authors should however test for interaction according to prior revascularization (15% of the MI population), as these individuals must have ischemic heart disease and thus a higher baseline risk of heart failure, or maybe even undiagnosed, pre-existing heart failure.

Response: Similar to the stratified cancer analysis the reviewer previously requested we've stratified our results on prior revascularization. However, and respectively so, in line with the previously mentioned cancer analyse we prefer not to formally test for interaction. Essentially, performing interaction test for all the model parameters included will likely result in (multivariate) very sparse data settings in which interaction tests are known to underperform. Furthermore, there is an issue of multiplicity of tests which will be only inflated by performing many interaction tests.

The following sections were changed to reflect these issues, page 10

“

“In addition to these sensitivity analyses, we performed a cancer- and revascularization-stratified analysis. Due to discrepancies in sample size between the subgroups we decided against formal interaction testing, which may suffer from inflated false positive rates and lower power in such settings[15]. “

Page 12

“Stratifying the sample on cancer diagnosis or history of revascularization showed broadly similar results between subgroups (See appendix table S3 and S4, focusing on the confidence intervals),

however precision was low due to the limited number of patients with an history of revascularization or cancer. “

Conclusion

- Given the large number of missing data for BMI, please omit conclusions about associations with weight.

Response: We fully agree with this and removed this from the conclusions.

Further

- Please change “gender” to “sex” throughout the paper, i.e. (Oxford dictionary)

Grammar

(in languages such as Latin, French, and German) each of the classes (typically masculine, feminine, common, neuter) of nouns and pronouns distinguished by the different inflections which they have and which they require in words syntactically associated with them. Grammatical gender is only very loosely associated with natural distinctions of sex.

Response: Done.

VERSION 3 – REVIEW

REVIEWER	Michelle Dalgas Skott Schmiegelow Department of Cardiology, Gentofte University Hospital, Herlev-Gentofte Hospital, Denmark
REVIEW RETURNED	06-Oct-2017

GENERAL COMMENTS	Thank you for your comments. Below is a few grammatical issues. Unfortunately the authors misunderstood my primary concern about the importance of time to reperfusion. If these data are not available (as suggested by Table 1), please add this to the limitations. Strengths and limitations: • Line 65: Myocardial infarctions patients• Line 67: Focused Results: • Line 228: 3,538 were (not where) Discussion: • Line 463: a history (not an)• Please add a discussion about the importance of time to reperfusion, preferably by adding a subanalysis including the 62.5% with available data (the authors misunderstood my previous comments). Further • Please be consistent and use either hospitalized and revascularisation, or hospitalized and revascularization.
--

VERSION 3 – AUTHOR RESPONSE

Reviewer: 3

Reviewer Name: Michelle Dalgas Skott Schmiegelow

Institution and Country: Department of Cardiology, Gentofte University Hospital, Herlev-Gentofte Hospital, Denmark

Please state any competing interests: None declared

Please leave your comments for the authors below

Thank you for your comments. Below is a few grammatical issues. Unfortunately the authors misunderstood my primary concern about the importance of time to reperfusion. If these data are not available (as suggested by Table 1), please add this to the limitations.

Strengths and limitations:

Response: Based on this comment and your suggestion below we have double checked our data and spotted a mistake in table 1 where the percentage missing reperfusion data should be 88.3% instead of the 37.5% previously reported. Given this large degree of missingness we feel any further analyses with this variable is likely unreliable and have decided against performing these. Please accept our sincere apology for this mix-up and the preceding misunderstanding.

We checked all data presented and corrected some remaining decimal errors in the percentage missing data of Table 1. These errors seem to stem for having to manually copy data from a secure portal into the tables, and are not due to mistakes in the analyses.

- Line 65: Myocardial infarctions patients

Response: Done

- Line 67: Focused

Response: Done

Results:

- Line 228: 3,538 were (not where)

Response: Thank you

Discussion:

- Line 463: a history (not an)

Response: Changed.

- Please add a discussion about the importance of time to reperfusion, preferably by adding a subanalysis including the 62.5% with available data (the authors misunderstood my previous comments).

Response: Please see our response above.

Further

- Please be consistent and use either hospitalized and revascularisation, or hospitalized and revascularization.

Response: Done

VERSION 4 – REVIEW

REVIEWER	Michelle Schmiegelow Department of Cardiology Gentofte Hospital Kildegaardsvej 28 2900 Hellerup, Denmark
REVIEW RETURNED	24-Oct-2017

GENERAL COMMENTS	The percentage of patients with no data on time to revascularization (surprisingly) has been changed from 37.5% to 88.3% and consequently makes analyses of the associations between time-to-revascularization and risk of heart failure unreliable. In STEMI-patients, prompt revascularization is of high priority in the modern era to reduce infarct size, the primary predictor of post-infarction heart failure.(see references below) I therefore find it crucial for the authors to add the lack of analyses regarding time to revascularization as a limitation. Stone GW, Selker HP, Thiele H, Patel MR, Udelson JE, Ohman EM, Maehara A, Eitel I, Granger CB, Jenkins PL, Nichols M, Ben-Yehuda O. Relationship between infarct size and outcomes following primary PCI: patient-level analysis from 10 randomized trials. J Am Coll Cardiol 2016;67(14):1674–1683. Ibanez B, Heusch G, Ovize M, Van de Werf F. Evolving therapies for myocardial ischemia/reperfusion injury. J Am Coll Cardiol 2015;65(14):1454–1471. 2017 ESC Guidelines for the management of acute myocardial infarction in patients presenting with ST-segment elevation: The Task Force for the management of acute myocardial infarction in patients presenting with ST-segment elevation of the European Society of Cardiology (ESC) Borja Ibanez Stefan James Stefan Agewall Manuel J. Antunes Chiara Bucciarelli-Ducci Héctor Bueno Alida L. P. Caforio Filippo Crea John A. Goudevenos Sigrun Halvorsen ... Show more European Heart Journal, ehx393, https://doi.org/10.1093/eurheartj/ehx393 Published: 26 August 2017
---

VERSION 4 – AUTHOR RESPONSE

Reviewer: 3

Reviewer Name: Michelle Schmiegelow

Institution and Country: Department of Cardiology, Gentofte Hospital, Kildegaardsvej 28, 2900 Hellerup, Denmark

Please state any competing interests: None declared

Please leave your comments for the authors below

The percentage of patients with no data on time to revascularization (surprisingly) has been changed from 37.5% to 88.3% and consequently makes analyses of the associations between time-to-revascularization and risk of heart failure unreliable. In STEMI-patients, prompt revascularization is of high priority in the modern era to reduce infarct size, the primary predictor of post-infarction heart failure.(see references below) I therefore find it crucial for the authors to add the lack of analyses regarding time to revascularization as a limitation.

Stone GW, Selker HP, Thiele H, Patel MR, Udelson JE, Ohman EM, Maehara A, Eitel I, Granger CB, Jenkins PL, Nichols M, Ben-Yehuda O. Relationship between infarct size and outcomes following primary PCI: patient-level analysis from 10 randomized trials. J Am Coll Cardiol 2016;67(14):1674–1683.

Ibanez B, Heusch G, Ovize M, Van de Werf F. Evolving therapies for myocardial ischemia/reperfusion injury. J Am Coll Cardiol 2015;65(14):1454–1471.

2017 ESC Guidelines for the management of acute myocardial infarction in patients presenting with ST-segment elevation: The Task Force for the management of acute myocardial infarction in patients presenting with ST-segment elevation of the European Society of Cardiology (ESC)
 Borja Ibanez Stefan James Stefan Agewall Manuel J. Antunes Chiara Bucciarelli-Ducci Héctor Bueno Alida L. P. Caforio Filippo Crea John A. Goudevenos Sigrun Halvorsen ... Show more
 European Heart Journal, ehx393, <https://doi.org/10.1093/eurheartj/ehx393>
 Published: 26 August 2017

Response: As suggested this limitation was included in the “Strengths and limitations of this study” section:

“Due to the high degree of missing data on time to revascularization (88.3%) we did not explore its relation with HF incidence.”

Additional we acknowledge this omission in the discussion

Page 12 “Related, we acknowledge we did not assess all potential predictors of post MI HF, for example due to the high degree of missingness we did not explore the prognostic potential of time to revascularization[30].“

VERSION 5 – REVIEW

REVIEWER	Michelle Dalgas Skott Schmiegelow Department of Cardiology, Herlev-Gentofte Hospital Gentofte Hospital Kildegaardsvej 28 2900 Hellerup Denmark
REVIEW RETURNED	08-Nov-2017
GENERAL COMMENTS	I acknowledge that the authors have added the limitation of missing information regarding time to revascularization to the paper including a reference, but I find it crucial to know if all the patients were revascularized?

VERSION 5 – AUTHOR RESPONSE

Reviewer: 3

Reviewer Name: Michelle Schmiegelow

Institution and Country: Department of Cardiology, Gentofte Hospital, Kildegaardsvej 28, 2900 Hellerup, Denmark

Please state any competing interests: None declared

Please leave your comments for the authors below

Comment: I acknowledge that the authors have added the limitation of missing information regarding time to revascularization to the paper including a reference, but I find it crucial to know if all the patients were revascularized?

Response: We acknowledge that the use of revascularization in England is an interesting research topic, and as we describe in the discussion, revascularization is indeed a relevant prognostic variable for post MI HF. Apparently these questions are of particular interest to the current reviewer, regrettably however, these questions were not the primary concern of our study which focussed on exploring the prognostic value of routinely collected healthcare data in predicting post MI HF.

Furthermore, as we explained before, we do not think that the current linked data can be usefully extended to provide valid answers to these questions. As presented in Table 1, information on revascularization is only available for a subset of our sample. This lack of data on revascularization is due to our focus on both primary care records and hospital records. While the latter source has detailed “Myocardial Ischaemia National Audit Project” (MINAP) data, the former does not. As such we decided against exploring the prognostic potential of (time to) revascularization.

Focussing on patients (included between 1998 to 2010) with a recorded revascularization procedure (Table 1) we note that only 19.3% of the STEMI patients were treated by a primary PCI. This is related to the relatively slower uptake of effective STEMI treatments, such as primary PCI, in England compared to other European countries (PMID: 23608389 & PMID: 24461715).

Despite this lack of information on revascularization, we feel our study provides a valid description of the utility of routinely collected healthcare data in predicting post MI HF. Likely in future electronic healthcare records will further mature, and revascularization as well as other important clinical characteristics, will be better recorded. At such a time, it would be interesting to update the presented results and determine how much these variables add. In general, routinely collected data are an opportune source to provide dynamic and up to date information on disease prognostication, which will challenge the current time-invariant view.

VERSION 6 – REVIEW

REVIEWER	Michelle Dalgas Skøtt Schmiegelow Department of Cardiology Gentofte Hospital Denmark
REVIEW RETURNED	12-Dec-2017

GENERAL COMMENTS	Are all patients revascularized?
----------------------------------

VERSION 6 – AUTHOR RESPONSE

Reviewer: 3

Reviewer Name: Michelle Schmiegelow

Institution and Country: Department of Cardiology, Gentofte Hospital, Kildegaardsvej 28, 2900 Hellerup, Denmark

Please state any competing interests: None declared

Please leave your comments for the authors below

I acknowledge that the authors have added the limitation of missing information regarding time to revascularization to the paper including a reference, but I find it crucial to know if all the patients were revascularized?

Response: We acknowledge that the use of revascularization in England is an interesting research topic, and as we describe in the discussion, revascularization is indeed a relevant prognostic variable for post MI HF. Apparently these questions are of particular interest to the current reviewer, regrettably however, these questions were not the primary concern of our study which focussed on exploring the prognostic value of routinely collected healthcare data in predicting post MI HF. Furthermore, as we explained before, we do not think that the current linked data can be usefully extended to provide valid answers to these questions. As presented in Table 1, information on revascularization is only available for a subset of our sample. This lack of data on revascularization is due to our focus on both primary care records and hospital records. While the latter source has detailed “Myocardial Ischaemia National Audit Project” (MINAP) data, the former does not. As such we decided against exploring the prognostic potential of (time to) revascularization.

Focussing on patients (included between 1998 to 2010) with a recorded revascularization procedure (Table 1) we note that only 19.3% of the STEMI patients were treated by a primary PCI. This is related to the relatively slower uptake of effective STEMI treatments, such as primary PCI, in England compared to other European countries (PMID: 23608389 & PMID: 24461715).

Despite this lack of information on revascularization, we feel our study provides a valid description of the utility of routinely collected healthcare data in predicting post MI HF. Likely in future electronic healthcare records will further mature, and revascularization as well as other important clinical characteristics, will be better recorded. At such a time, it would be interesting to update the presented results and determine how much these variables add. In general, routinely collected data are an opportune source to provide dynamic and up to date information on disease prognostication, which will challenge the current time-invariant view.